# Structural basis of leukotriene B4 receptor 1 activation

Na Wang[1], Xinheng He [2,3], Jing Zhao[4], Hualiang Jiang [2,3], Xi Cheng [2,3], Yu Xia[4], H. Eric Xu [2,3] & Yuanzheng He [1✉]

Leukotriene B4 receptor 1 (BLT1) plays crucial roles in the acute inflammatory responses and is a valuable target for anti-inflammation treatment, however, the mechanism by which leukotriene B4 (LTB4) activates receptor remains unclear. Here, we report the cryo-electron microscopy (cryo-EM) structure of the LTB4 -bound human BLT1 in complex with a $G_i$ protein in an active conformation at resolution of 2.91 Å. In combination of molecule dynamics (MD) simulation, docking and site-directed mutagenesis, our structure reveals that a hydrogen-bond network of water molecules and key polar residues is the key molecular determinant for LTB4 binding. We also find that the displacement of residues M101[3.36] and I271[7.39] to the center of receptor, which unlock the ion lock of the lower part of pocket, is the key mechanism of receptor activation. In addition, we reveal a binding site of phosphatidylinositol (PI) and discover that the widely open ligand binding pocket may contribute the lack of specificity and efficacy for current BLT1-targeting drug design. Taken together, our structural analysis provides a scaffold for understanding BLT1 activation and a rational basis for designing anti-leukotriene drugs.

[1] Laboratory of Receptor Structure and Signaling, The HIT Center for Life Sciences, Harbin Institute of Technology, Harbin 150001, China. [2] The CAS Key Laboratory of Receptor Research and State Key Laboratory of Drug Research, Shanghai Institute of Materia Medica, Chinese Academy of Sciences, Shanghai, China. [3] University of Chinese Academy of Sciences, Beijing, China. [4] MOE Key Laboratory of Bioorganic Phosphorus Chemistry & Chemical Biology, Department of Chemistry, Tsinghua University, Beijing 100084, China. ✉email: ajian.he@hit.edu.cn

LTB4 is an inflammatory lipid mediator produced from arachidonic acid, and is one of the most important leuko-trienes in the onset of acute inflammatory responses[1]. LTB4 acts as a chemoattractant for neutrophils and macrophages to vascular endothelium[2]. The binding of LTB4 to its receptors (BLT1 and BLT2) activates leukocytes and prolongs their survival[3]. BLT1 expresses exclusively in the immune system while BLT2 has a broad spectrum of expression profile. In general, BLT1 has a higher affinity for LTB4 than BLT2[4]. On the other side the heptadecanoid 12(S)-hydroxyheptade catrienoic acid (12-HHT) has been discovered as the endogenous ligand for BLT2, which has similar affinity to BLT2 as LTB4 to BLT1[5]. 12-HHT is an enzymatic product of thromboxane $A_2$ (TXA$_2$) synthase in the conversion of prostaglandin H 2 (PGH$_2$) to TXA$_2$, 12-HHT and malondialdehyde. A major difference between 12-HHT and LTB4 chemical structure is that the 12-HHT only has a hydroxyl group at the C12 position, while LTB4 has two hydroxyl groups, one at the C12 position and one at the C5 position. Whether this dif-ference determines the specificity of LTB1 and LTB2 is an interesting question to ask. The LTB4-BLT1 signaling pathway is crucial for many inflammatory diseases, such as infectious dis-eases, allergy, autoimmune diseases, and metabolic disease[6]. Emerging evidence also suggested a complex role of the LTB4-BLT1 signaling in cancers. Comparing to cysteinyl leukotriene (CysLTs) which mainly act in the airway, LTB4 is regarding as one of the most potent neutrophil chemoattractants in the host defense system against infection and invasion from foreign bodies[1,6].

BLT1 and BLT2 belong to the class A G-protein coupled receptors (GPCRs), and can be further classified as members of lipid receptor subfamily which includes prostanoid receptors, sphingosine receptors, lysophospholipid receptors, cannabinoid receptors etc. BLT1 and BLT2 are mainly coupled with G$_i$ and G$_q$ pathways. BLT1 is an attractive drug target for allergic airway inflammation, inflammatory arthritis, atherosclerosis, and psoriasis[3]. Despite extensive efforts in developing BLT-targeting compounds, none of them have yet made to the market due to the high incidence of side effects, low specificity, and poor efficacy. For instance, etalocib (LY293111), has been demonstrated to activate peroxisome proliferator activator receptor gamma (PPARγ)[7]. Part of the reason for the unsuccessful drug design may be due to the limited structural information of BLT available, particularly, the mechanistic insights into receptor activation. Currently, there are only two antagonist-bound BLT1 receptor structures, one is BIIL260-bound guinea pig BLT1[8], and the other is MK-D-046-bound human BLT1[9]. The absence of agonist-bound BLT1 structures has limited mechanistic understanding of the ligand recognition and receptor activation. Here, we report the cryo-EM structure of the endogenous ligand LTB4-bound human BLT1 in complex with G$_i$ protein in an active con-formation. Together with MD simulation, docking, and muta-tional studies, the structural information of our study provides a framework for understanding LTB4 signaling and a rational basis for designing novel anti-leukotriene drugs.

## Results

**The overall structure of BLT1/Gi complex**. To facilitate receptor expression and complex assembling, we use an engineered receptor, which contains 4 thermostabilizing mutations (L106$^{3.41}$W, A196$^{5.53}$I, C287$^{7.55}$F, and S310A, superscripts refer to the Ballesteros-Weinstein numbering) adopted from the MK-D-046-bound human BLT1 crystal structure (Supplementary Fig. 1a and methods for details)[9]. The original design of the thermostabilizing mutations contains 5 point mutations, includ-ing an extra S116$^{3.51}$Y mutation in addition to the 4 mutations

mentioned above. We excluded the S116$^{3.51}$Y mutation which may affect receptor activation as suggested by the initial report. We use the serum response element (SRE) reporter assay[10], an established reporter assay for G$_i$ signaling (Supplementary Fig. 1b and methods) to examine the activity of the mutants. The 4-mutation construct shows enhanced receptor activity which may be due to the enhanced stability of the receptor. To our surprise, the 5-mutation which has the S116$^{3.51}$Y mutation at the DRY motif also shows enhanced receptor activity, although slightly lower than the 4 mutations. In addition, we use a NanoBiT tethering strategy[11] in which the C-terminus of BLT1 was fused to the large part of NanoBiT (LgBiT), and the C-terminus of Gβ was fused to the renovated 13-amino acid peptide of NanoBiT (HiBiT). The introduction of the LgBiT to receptor do not affect the overall receptor activation is evidenced by the similar transactivation activity compared with the wild-type receptor (Supplementary Fig. 1b). A dominant-negative mutation of Gα$_{i1}$[12] together with the scFv16 antibody[13] was used for complex assembling. The complex was purified from Sf9 cells co-infected with baculoviruses encoding each component of the complex (Supplementary Fig. 1c and methods). The complex structure was solved by the single-particle cryo-EM analysis at a resolution of 2.91 Å according to the gold standard of FSC = 0.143 (Supplementary Figs. 2,4 and methods). Both the receptor side and the G-protein side have a clean and clear density, the LTB4 ligand is well resolved in the ligand-binding pocket. A phospholipid inositol (PI) molecule was found at the membrane side (Fig. 1). In addition, 4 cholesteryl hemisuccinate (CHS) molecules were found to surround the TMD of BLT1 (Fig. 1). Local resolution analysis shows that the core of receptor trans-membrane domain and the Gβ subunit have the highest resolu-tion, the extracellular part of the receptor, the border of Gα RAS domain and the N-terminus of Gβγ have the relatively lower resolution (Supplementary Fig. 3a). We did not observe a clear density for the alpha-helical domain of the Gα$_i$ subunit, similar to most GPCR-G protein complex structures[14]. The overall elec-tronic density of the receptor is of high quality and we are able to resolve almost the whole receptor except for the first 14 residues of the N-terminus and the C-tail after residue 301. The archi-tecture of the BLT1/G$_i$ complex resembles most GPCR/G-protein complex, in which the G-protein uses its distal end of the α 5 helix (αH5) of the Gα subunit as a main interface to interact with the open intracellular core of the receptor (Fig. 1).

**The ligand-binding pocket of BLT1**. The native ligand LTB4 is well resolved in the ligand-binding pocket of BLT1 (Fig. 2a) as it binds the receptor in an extended conformation with the car-boxylate in the aqueous environment at the entrance to the orthosteric binding pocket. The pocket is mainly formed by a cluster of hydrophobic residues from transmembrane helices (TM-2,3,6 and 7), including F77$^{2.63}$, L78$^{2.64}$, F74$^{2.60}$, M101$^{3.36}$, Y102$^{3.37}$, W234$^{6.48}$, Y237$^{6.51}$, F275$^{7.43}$, and I271$^{7.39}$ (Fig. 2a,b). Those residues form a highly hydrophobic tunnel in the pocket which matches the lipid nature of LTB4 very well (Supplementary Fig. 5a, left panel). An electrostatic potential analysis shows that major part of LTB4 is neutral while the carboxyl head is negative charged, which match the receptor electrostatic potential well (Supplementary Fig. 5a, right panel). A few polar residues, including H94$^{3.29}$, R156$^{4.64}$, E185$^{5.42}$, H238$^{6.52}$, R267$^{7.35}$ and N268$^{7.36}$ also form the outer rim of the pocket. We identified a direct polar interaction between N268$^{7.36}$ and the C1 carboxyl group of LTB4, and a cluster of water molecules that connect LTB4 to the key polar residues of the receptor. Particularly, water molecule 1 (W1) forms a hydrogen bond network that connect R156$^{4.64}$ and R267$^{7.35}$ to the C5 hydroxyl group of LTB4 (Fig. 2a).

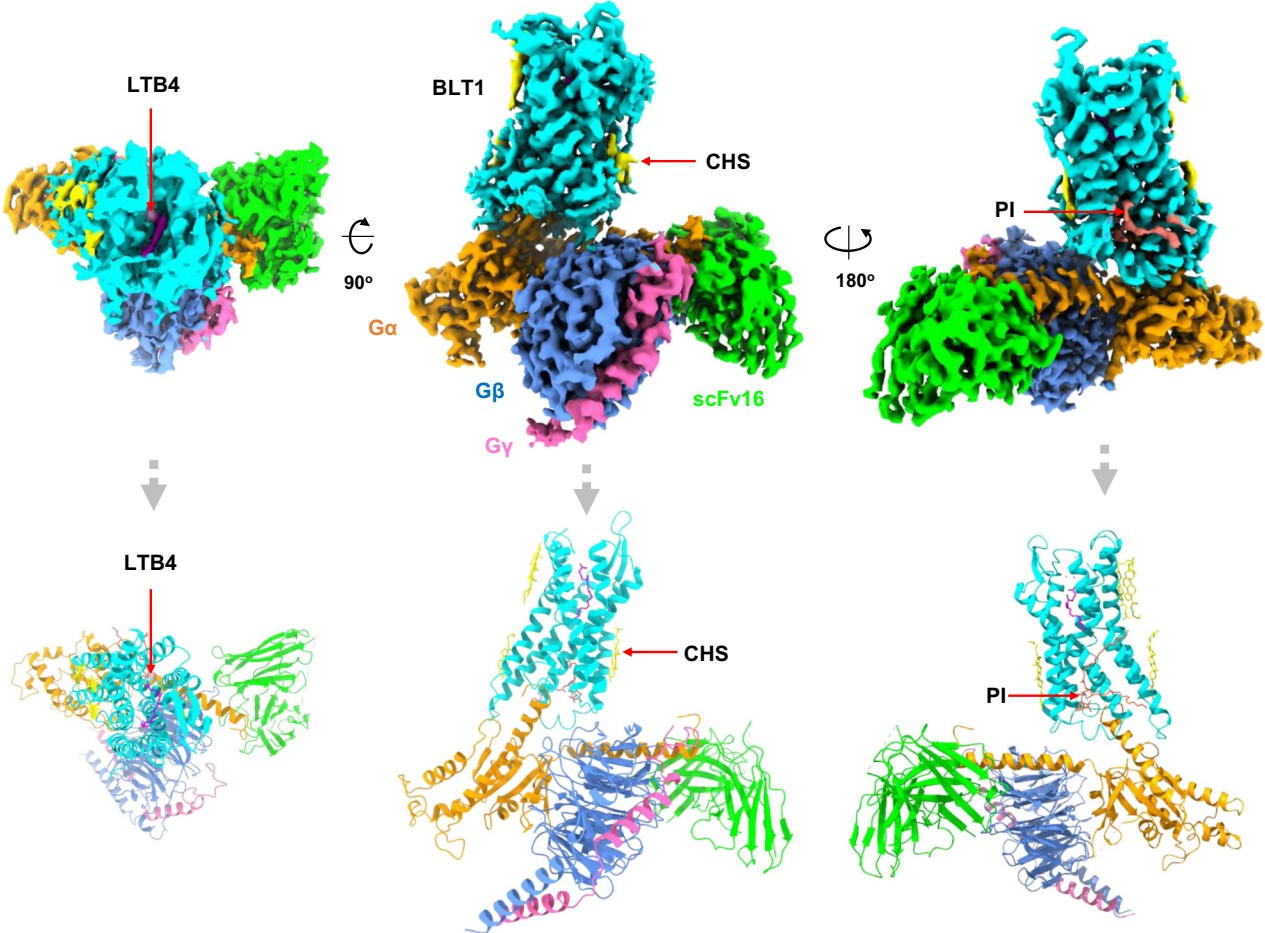

**Fig. 1 Overall structure of the BLT1/G$_i$ complex.** Upper panel, orthogonal views of the cryo-EM density map of the BLT1/G$_i$ complex; lower panel, model of the complex in the same view and color scheme as shown in the middle-upper panel.

R267[7.35] also forms a hydrogen bond interaction with water molecule 2 (W2). In addition, water molecule 6 interacts with N268[7.36] and is connected to water molecule 7. We also measure the distance between those water molecules and the connected LTB4 and the key residues of the receptor. The distances vary from 3.0 Å to 3.3 Å, in a good range to form ideal hydrogen bond, therefore these densities are highly likely to be water molecules that connect LTB4 to the receptor. Together, these water molecules form a hydrogen bond network that connects polar residue N268[7.36], R156[4.64], R267[7.35], and H94[3.29] to LTB4 to lock the ligand in the pocket. We use MD simulations to examine the hypothesis. If the ligand-binding pocket is energy unfavorable, the ligand LTB4 will quickly jump out of the pocket. The 200 ns triplicated runs all show that the LTB4 molecule sticks within the pocket, in poses very similar to the original LTB4 position we observed in our cryo-EM structure, throughout whole simulations (Fig. 2c and Supplementary Movie 1). In the simulations, we also observed a stable association between H94[3.29] and the C5 hydroxyl group of LTB4 right after the start of the simulation and a very dynamic water hydrogen network that connects LTB4 to the key polar residue R156[4.64], R267[7.35], H94[3.29], N268[7.36], and E185[5.42] (Supplementary Fig. 5b). We further use docking methods to validate our ligand position in the pocket. The docking results show that LTB4 docks in the position very similar to that we observed in our cryo-EM structure with or without the existence of the water molecules in the pocket. However, with the 5 water molecules, the docking score is lower than without water (−8.54 vs −8.20, Supplementary Fig. 5c).

We use an SRE-reporter assay to examine the contributions of those residues in the pocket. The data shows that F74[2.60]A, L78[2.64]F, H94[3.29]A, M101[3.36]A, E185[5.42]A, W234[6.48]A, I271[7.39]A, and F275[7.43]A cause a big decrease of receptor activation, and Y102[3.37]A, R156[4.64]A, and G189[5.46]F have the most detrimental effects, while F77[2.63]A, H238[6.52]A, and N268[7.36]A do not affect receptor's activity. We are not totally surprised that the N268[7.36]A has no effect on receptor activity as it directly interacts with the carboxyl end of LTB4 (Fig. 2a). Our MD simulation data shows that the carboxyl end of LTB4 has certain degree of freedom (Supplementary Movie 1), allowing it to form hydrogen bonds with the backbone of TM7 and a water-mediated network that connects it to the receptor (Supplementary Fig. 5b). More importantly, the carboxyl end of LTB4 modified by the fluorophore Alexa Fluor 568 shows a full agonist activity[15], indicating the N268[7.36]/carboxyl interaction is additive, not determinant. The large decrease of activities by those bulky side chain to small side chain mutants (F74[2.60]A, Y102[3.37]A, M101[3.36]A, W234[6.48]A, I271[7.39]A, and F275[7.43]A), is consistent with their roles in forming the skeleton of the hydrophobic tunnel in the pocket. Conversely, the great decrease of receptor activity by the "small to large" mutants (L78[2.64]F and G189[5.46]F) is consistent with their roles in forming the beginning and the ending of the pocket, as the bulky side chain of phenylalanine may block the entry or disturb the docking pose of LTB4 in the pocket. Interestingly, the dramatic decrease of activity is seen on the polar mutant R156[4.64]A, R267[7.35]A, and H94[3.29]A, which almost completely abrogate the receptor activity. These data are

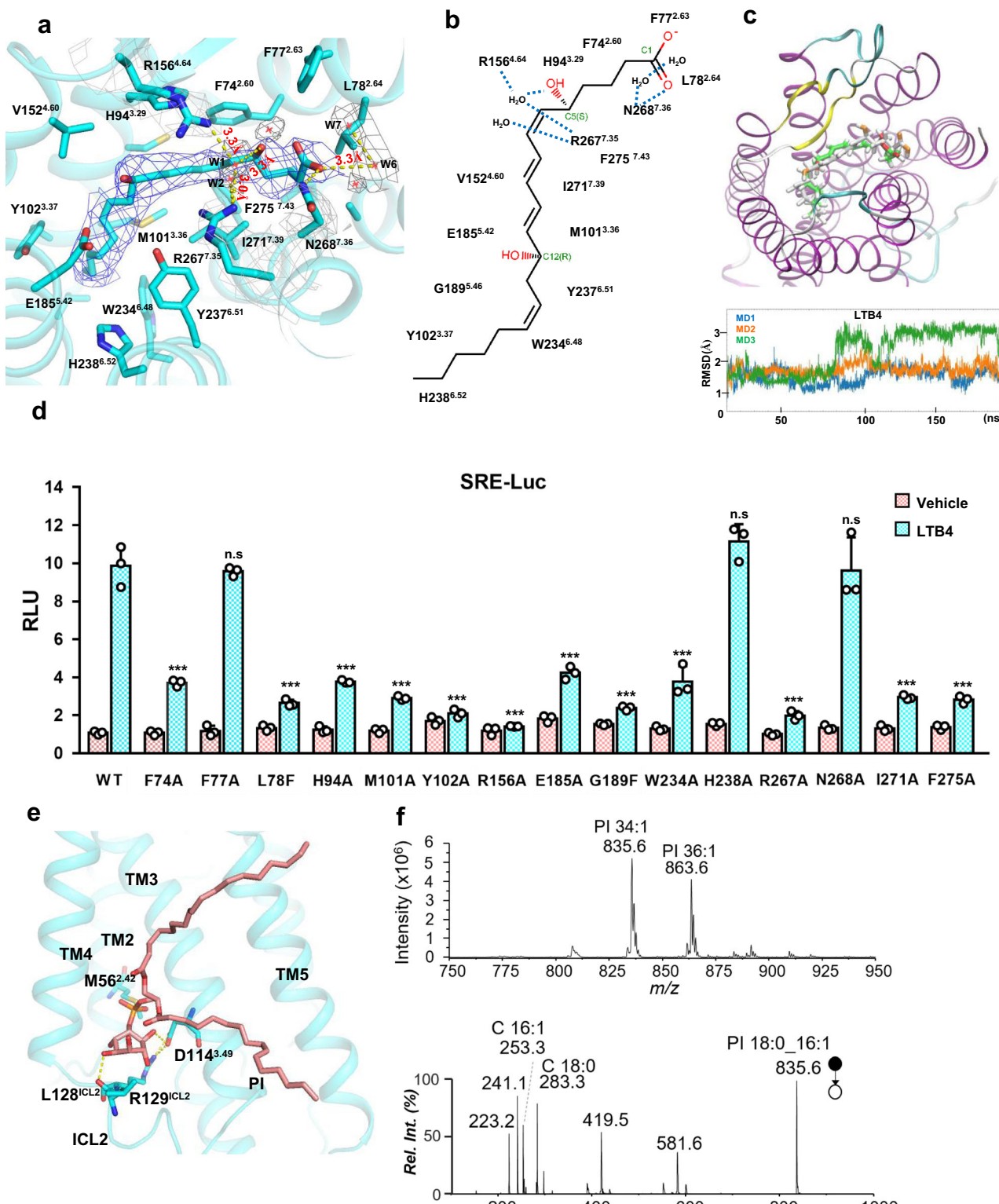

consistent with our structural observation and the MD simulation outcome. The mutation data of R156[4.64]A and H94[3.29]A is also consistent with a previous study by Basu et al.[16].

**PI binding site**. Lipids play important roles in regulating the activity of membrane protein, particularly phospholipids have been demonstrated to play a key role in regulating the activity of GPCR[17,18]. Interestingly, we identified a cluster of density that

resembles a PI molecule in the middle of TM3, TM4, and intracellular loop 2 (ICL2), facing the membrane side (Fig. 1, Fig. 2e). We used mass spectrometry to identify the lipid component of our complex. Compared to the buffer employed in purifying the receptor/G-protein complex, we identified two clear PI peaks at $m/z$ 835.6 and 863.6 ([M − H]$^-$, Fig. 2f) after liquid chromatography-mass spectrometry analysis. Further tandem mass spectrometry analysis of these two peaks provides unequivocal evidence that they are PI 18:0/16:1 and PI 18:0/18:1,

**Fig. 2 Ligand binding pocket of BLT1. a** BLT1 ligand binding pocket for LTB4. The LTB4 molecule is shown in magenta in the middle of the pocket, density map of LTB4 (blue mesh) is set at contour level of 5.0, density map of water (grey mesh) is set at contour level of 4.0, density map of R156[4.64] and R267[7.35] is set at contour level of 4.0. Surrounding residues within 5.0 Å of LTB4 are shown in sticks and colored in cyan. **b** The scheme of the chemical structure of LTB4 and surround residues. **c** A representative snapshot of MD simulation. LTB colored in white is the original pose in the cryo-EM structure, LTB colored in green is the representative pose of the MD simulation. Lower panel, RMSD of the LTB4 trajectories in the simulations. **d** A SRE reporter assay of key mutants in the ligand-binding pocket. LTB4, 300 nM; data are presented as mean values ± SD; $n = 3$ independent samples; n.s. no significant; *$p < 0.05$; **$p < 0.01$; ***$p < 0.001$. The exact $p$ value for F74A, F77A, L78A, H94A, M101A, Y102A, R156A, E185A, G189F, W234A, H238A, R267A, N268A, I271A, F275A are: 0.0005, 0.66, 0.0003, 0.0005, 0.0003, 0.0002, 0.0001, 0.0009, 0.0002, 0.001, 0.19, 0.0002, 0.84, 0.0003, 0.0003, respectively. T-test, two tailed, sample equal variance. RLU, relative luciferase unit. WT, wild-type. **e** Detail of the PI binding pocket. **f** Mass spectrometry analysis of lipid component of the BLT1/$G_i$ complex.

respectively (Fig. 2f and Supplementary Fig. 6a,b). Because of the relative weak density on the alkyl chain, we cannot distinguish PI 18:0/16:1 and PI 18:0/18:1 from the map, therefore we modeled PI 18:0/16:1 into the density given its higher relative ion abundance in the mass spectrometry analysis (Supplementary Fig. 6c, left panel). The model shows that the PI head group inserts into a cavity formed by TM2, TM3, TM4, and ICL2. Particularly, the hydroxyl group of the C2 position of the inositol ring forms a hydrogen bond network with D114[3.49] and R129[ICL2], and the hydroxyl group of the C5 position of the inositol ring forms hydrogen bond interaction with the backbone carbonyl of L128[ICL2] (Fig. 2e). On the other side, residue M56[2.42] forms a hydrophobic wall of the cavity, blocking the further insertion of the PI head group into the receptor core. Mutations of D114[3.49]A, R129[ICL2]A and M56[2.42]K all cause large decreases of receptor activity (Supplementary Fig. 6d). A comparison of the D114[3.49] site in the lipid receptor family shows that the site is conserved in the lipid receptor family except for the CysLTs (Supplementary Fig. 6e). To rule out that PI 18:0/18:1, the less abundant PI molecule identified in the LC-MS analysis, may have a different conformation in the pocket, we also modeled it in the pocket. The data shows that the head group of PI 18:0/18:1 adopts almost identical conformation as PI 18:0/16:1 (Supplementary Fig. 6c, right panel), while the only difference is the length of the alky chain. Recently, a phospholipid phosphatidylinositol 4-phosphate (PtdIns4P) allosteric binding site was found in the cryo-EM structures of serotonin receptor 5-HT$_{1A}$/$G_i$ complex[18]. A comparison of our PI binding site with the PtdIns4P binding sites shows the significant difference between 5-HT$_{1A}$ and BLT1. In 5-HT$_{1A}$, the PtdIns4P binding site lies between TM6, TM7, and H8; in BLT1, the PI binding site lies between TM3, TM4, TM5, and ICL2, in a totally opposite direction as in 5-HT$_{1A}$ (Supplementary Fig. 6e). The function of the PI binding site is unknown, however, we suspect other molecules may also utilize this site to regulate receptor activation in the real membrane environment. It is known that phosphatidylinositol 3-kinase (PI3-K) activation via LTB4/BLT1 induced Gi signaling is a pre-requirement for enzyme release in leukemia cells[19], and PI3Kβ plays a critical role in neutrophil migration via LTB4/BLT1 activation[20]. We therefore first modeled a PtdIns4P molecule into a pocket, the data shows that a PtdIns4P molecule can be well set in the pocket. We also successfully modeled a phosphatidylethanolamine (PE 16:0/16:0) molecule, a component of the membrane, into the pocket. Interestingly, we found that the amine head group of the PE molecule inserted into the same tunnel previously discovered by the MK-D-046-bound BLT1 crystal structure[9], where a nonaethylene glycol (P2E) molecule snuggled in (Supplementary Fig. 6g). These data show the versatility of this binding site.

**Active vs inactive**. As seen in many GPCRs, the biggest difference between the inactive and active receptor is the outward displacement of TM6 by the receptor activation. In addition, a small inward displacement of TM7 was observed in the active

conformation (Fig. 3a). A comparison of the size of the ligand pocket shows that there is almost no change between agonist LTB4 and antagonist MK-D-046 binding (Supplementary Fig. 7a). Looking into the ligand-binding pocket, we noticed that agonist and antagonist use different strategies for receptor binding. For agonist, the alkyl chain of LTB4 inserts into a tunnel formed by TM3, TM4 and TM5, while for antagonists, both MK-D-046 and BIIL260 utilize the center tunnel of the receptor to accommodate their bulky tails (Supplementary Fig. 7b), particularly, the BIIL260 compound's benzamidine tail penetrates deeply into the middle tunnel of the receptor[8,9]. A close look at the ligand-binding pocket revealed that the most notable difference is the pose of M101[3.36] between the agonist binding and antagonist binding (Fig. 3b–d). In LTB4 binding, there is a 50° rotation of the side chain of M101[3.36] toward the center of the receptor to accommodate the alkyl tail of LTB4 where is the position previously occupied by that of M101[3.36] in the antagonist-bound receptor. We also observed notable difference in I271[7.39], in agonist binding, the side chain of I271[7.39] is pointed to the middle of the receptor (Fig. 3d); while in antagonists binding, I271[7.39] adopts an upward displacement. Both M101[3.36]A and I271[7.39]A mutants show significantly loss of the receptor activity (Fig. 2d). A key difference between the agonist and antagonist binding is direct and indirect polar interaction. In antagonist MK-D-046 binding, R156[4.64] forms hydrogen bonds with the carbonyl sulfonamide group of MK-D-046 and H94[3.29] forms hydrogen bonds with both the carbonyl sulfonamide and the hydroxyl group of the chromanol core of MK-D-046 (Fig. 3b). Whereas agonist LTB4 binds with R156[4.64] and H94[3.29] via water molecules in a hydrogen bond network (Fig. 2a and Supplementary Fig. 5b). Similarly, the antagonist BIIL260 binding also involves a rotation of H94[3.29] to form direct hydrogen bond with the middle ester group of BIIL260 (Fig. 3c). These antagonists penetrate directly into the deep hole of lower part in the center of seven transmembrane bundle with polar interaction with the receptor to keep in inactive conformation. In MK-D-046 binding, a sodium ion makes a network of polar interaction with S104[3.39], D64[2.50], S278[7.46] and S277[7.45] (Fig. 3b, lower part). The sodium ion has been demonstrated to play an important role to stabilize the inactive conformation of GPCRs among numerous studies[8,21]. While for the antagonist BIIL260 case, this interaction is mediated by the benzamidine group of BIIL260 (Fig. 3c, lower part). On the contrary, in agonist LTB4 binding, M101[3.36] swing into the center, with the inward side-chain rotation of I271[7.39] accompanied with the tilting of F275[7.43], to shut down the sodium ion to enter the lower part of pocket (Fig. 3d), thus allowing the receptor to unlock and rearrange the lower core for the αH5 of Gα$_i$ binding. To test this hypothesis, we mutated S277[7.45] and S104[3.39] to lysine, rendering their abilities to form salt bridge with D64[2.50] thus to permanently lock receptor in an inactive pose. Indeed, both S277[7.45]K and S104[3.39]K, alone or together, almost totally cripple receptor activation in the SRE reporter assay (Fig. 3e).

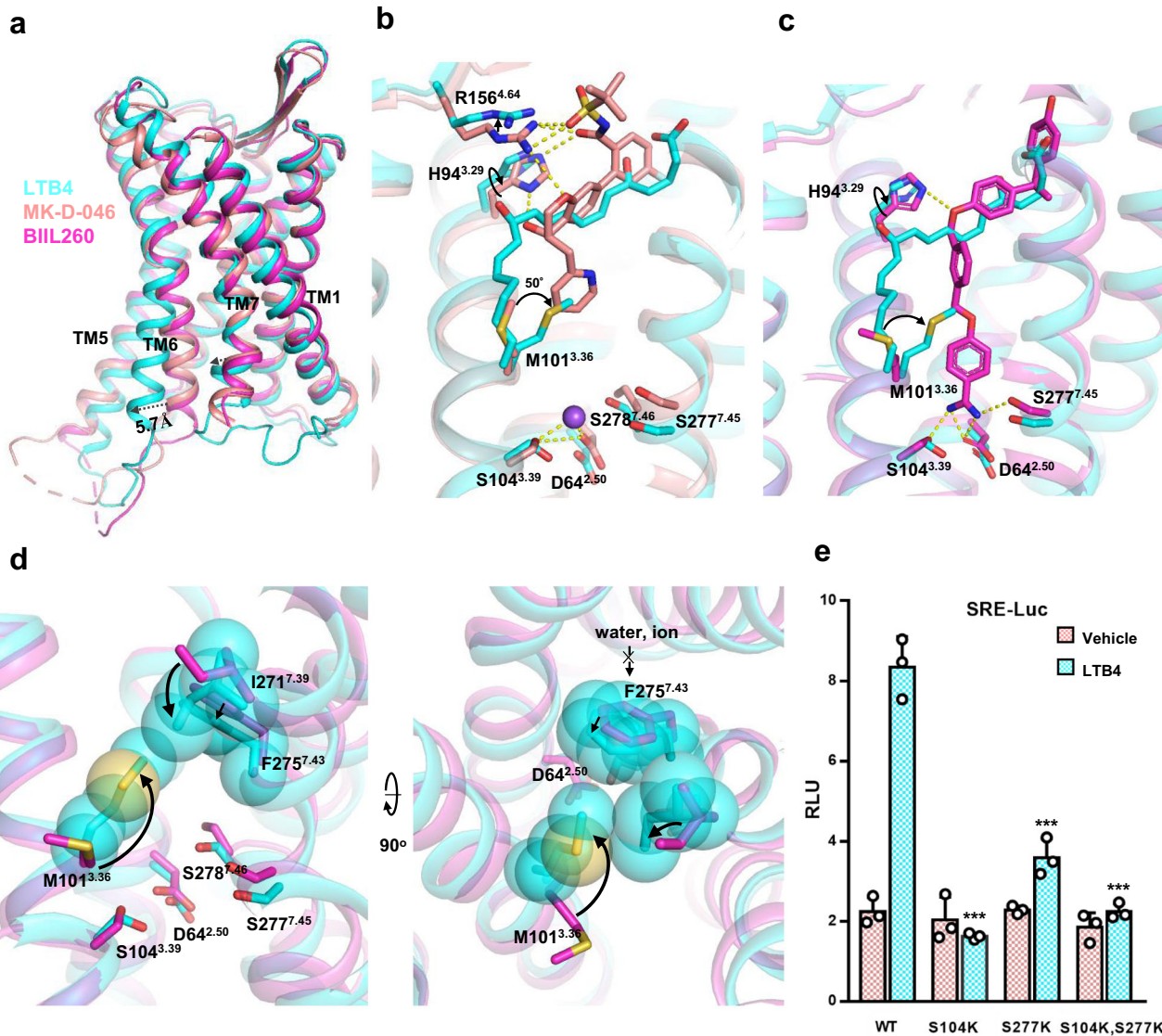

**Fig. 3 Comparison between inactive and active BLT1 structures. a** An overall comparison of LTB4-bound (cyan), MK-D-046-bound (brown) and BIIL260-bound (magenta) BLT1. A 5.7 Å outward displacement (dash arrow) of TM6 was measured by the Cα of R221[6.35]. **b** A detailed comparison of polar interaction between the LTB4-bound and the MK-D-046-bound receptor (PDB:7k15). **c** A detailed comparison of polar interaction between the LTB4-bound and the BIIL260-bound receptor (PDB:5×33). **d** The displacements of key residues that block the entry of water or ion molecules into the lower part of ligand binding pocket upon agonist LTB4 binding. **e** A SRE reporter assay of key mutants that lock receptor in the inactive conformation. LTB4, 300 nM; data are presented as mean values ± SD; $n = 3$ independent samples; n.s. no significant; $*p < 0.05$; $**p < 0.01$; $***p < 0.001$. The exact $p$ value for S104K, S277K, S104K/S277K are 0.0001, 0.0007, 0.0001, respectively. $T$-test, two tailed, sample equal variance.

We also studied the conserved PIF, NPxxY, and DRY motifs of the receptor core upon the receptor activation. In the PIF motif, F230[6.44] shows a significant displacement toward TM5 upon agonist binding (Supplementary Fig. 8a). In the NPxxY motif, Y285[7.53] shows a large displacement (~7 Å) toward the center of the receptor core (Supplementary Fig. 8b) and makes new contacts with M111[3.46] and S112[3.47] upon agonist binding. In the DRY motif, the agonist binding causes a sway of the sidechain of R115[3.50] toward the center of the cavity, but still interacts with Y201[5.58] on one side (Supplementary Fig. 8c) and on the other side directly interacts with the top of αH5 of Gαi. Beside the ligand-binding pocket and the conserved motif, the antagonist MK-D-046 bound receptor has an outward displacement of the extracellular side of TM7 and a sinking displacement of the ECL3 relative to the LTB4-bound receptor (Supplementary Fig. 8d).

**Gi engagement.** Resembling most Gi-coupled receptors, the engagement of Gi is mainly maintained by key interactions with BLT1 from TM3, TM5, TM6, TM7-H8 kink and ICL2[22]. The R218[6.32]/F354[G.H5.26] and R211[5.68]/D341[G.H5.13] interactions (Fig. 4a) have been seen many other Gi-coupled receptors, for example, dopamine receptor D2R[23], cannabinoid receptor CB1[24], and opioid receptor μOR[25]. As mentioned earlier, in the active conformation, the side chain of R115[3.50] of the DRY motif inserts into the center of the receptor make a direct hydrogen bond interaction with the backbone carbonyl of the C351[G.H5.23] of αH5, and this has also been seen in D2R[23] and neurotensin receptor NSTR1 Gi (Fig. 4b) interaction[26]. Interestingly, we also identify some distinct interaction between BLT1 and Gi. For instance, the G290[7.58] of the TM7-H8 kink makes a hydrogen bond interaction with the backbone carbonyl of L353[G.H5.25] of Gαi (Fig. 4a). In addition, R218[6.32] interacts with the backbone

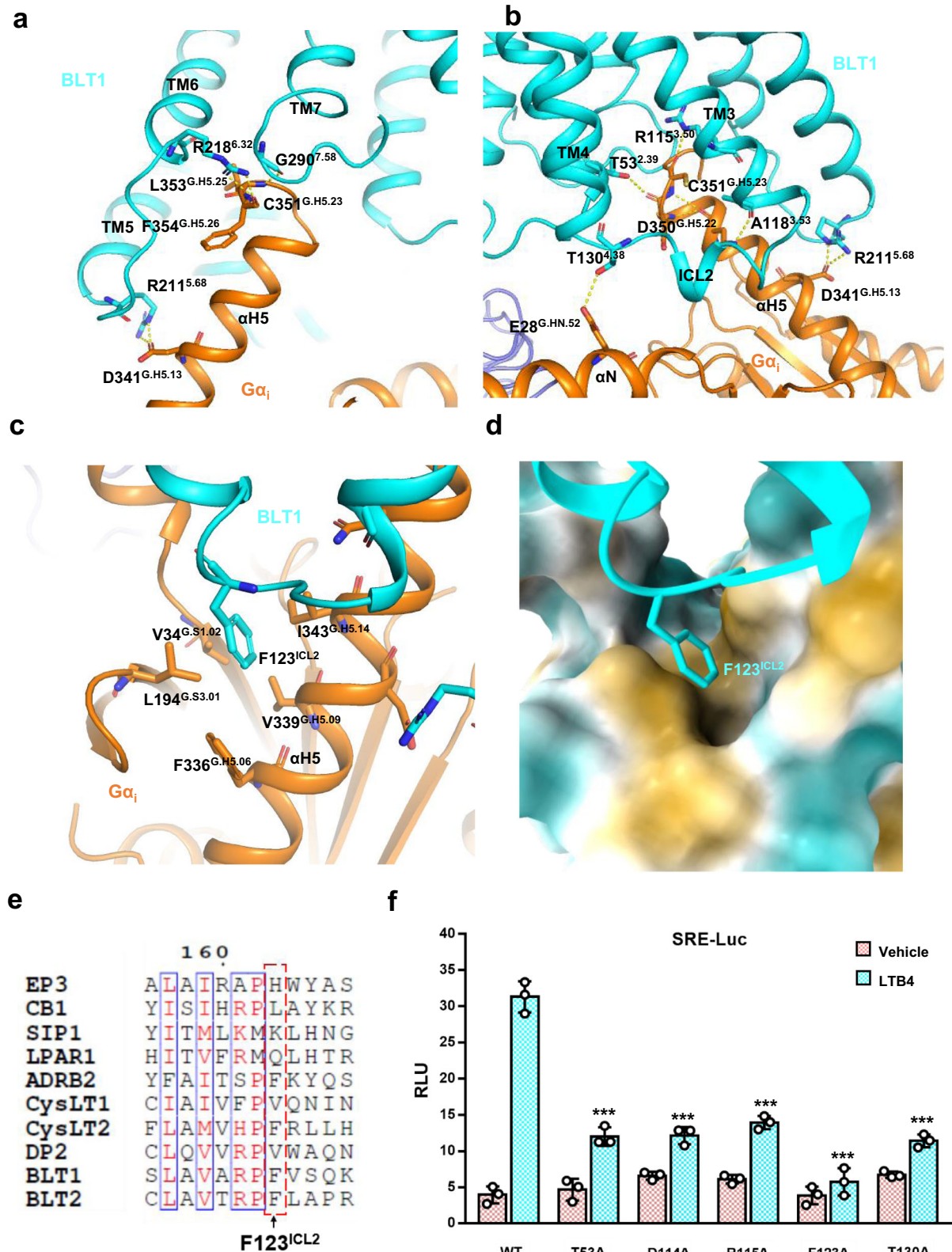

**Fig. 4 The Gi engagement of BLT1. a** The interaction between the receptor and the αH5, viewing from the TM5, TM6, TM7-H8 front angle. **b** The interaction between the receptor and Gαi, viewing from the TM3, TM4, and ICL2 front angle. **c** A detailed analysis of the F123$^{ICL2}$/Gα$_i$ interaction. **d** The insertion of F123$^{ICL2}$ into the hydrophobic cavity of Gα$_i$. The light orange color indicates hydrophobicity. **e** An alignment of BLT1 with lipid receptors around the F123$^{ICL2}$ region. **f** A SRE reporter assay to analyze the contribution of key residues in the G$_i$ engagement. LTB4, 300 nM; data are presented as mean values ± SD; $n = 3$ independent samples; n.s. no significant; *$p < 0.05$; **$p < 0.01$; ***$p < 0.001$. The exact $p$ value for T53A, D114A, R115A, F123A, T130A, are: 0.0001, 0.0001, 0.0002, 0.0001, 0.0001, respectively. $T$-test, two tailed, sample equal variance.

carbonyl of $C351^{G.H5.23}$, $T53^{2.39}$ interacts with the backbone carbonyl of $D350^{G.H5.22}$ of Gαi, and the $T130^{4.38}$ of ICL2 makes a direct interaction with the $E28^{G.HN.52}$ of the αN of Gαi (Fig. 4b). Most interestingly, the $F123^{ICL2}$ of ICL2 inserts into a hydrophobic cavity formed by $I343^{G.H5.14}$, $V339^{G.H5.09}$, $F336^{G.H5.06}$, $V34^{G.S1.02}$, and $L194^{G.S3.01}$ of the Gαi (Fig. 4c,d). An alignment of BLTs with CysLTs and members of the lipid receptor family, as well as β2 adrenergic receptor (β2AR), shows that the $F123^{ICL2}$ is conserved in BLT1 and BLT2. $F123^{ICL2}$ is also found in CysLT2 and β2AR, but not in lipid receptor LPAR1, S1PR1, CB1, EP3, DP2 (Fig. 4e). Importantly, mutations of the key residues in the $G_i$ interface decrease BLT1 activation, including $R115^{3.50}A$ of the DRY motif, and $T130^{4.38}A$, which destroys the ICL2/αN interaction, and $F123^{ICL2}A$ mutation which almost totally abrogates the receptor activation in the SRE reporter assay (Fig. 4f). In addition, we also did a comparison of the BLT1/$G_i$ complex with the $β_2AR/G_s$ complex (Supplementary Fig. 8e). The comparison shows that the TM6 of the $β_2AR/G_s$ complex has a more pronounced outward movement as seen in many $G_s$ complex, the comparison also shows that there are more polar interactions between TM5 and αH5 in the $β_2AR/G_s$ complex than in the BLT1/$G_i$ complex, and the head of the αH5 inserts deeper in the $β_2AR/G_s$ complex than in the BLT1/$G_i$ complex. We speculate those difference may account for the selectivity of $G_i$ for BLT1, and may also contribute to the higher stability of $G_s$ complexes seen in many cases.

**The upper open pocket of BLT1**. Compared to other lipid receptors, one unique feature of BLT1 is the widely open pocket on the extracellular side[8,9]. Most lipid receptors have a "cap" or a "lid" to cover the extracellular side of the ligand binding pocket to prevent non-specific binding and solvent entry. For instance, in CB1, both the N-terminal loop and the extracellular loop 2 (ECL2) pack against the center of the ligand-binding pocket (Supplementary Fig. 9a)[27], serve as the lid to prevent the entry of solvent molecule. In prostaglandin E receptor EP3, the long ECL2 mainly composed of two anti-parallel β-sheets serves as the lid to cover up the pocket[28]. In CysLT2, the ECL2 is the lid[29]. In sphingosine 1-phosphate receptor S1PR1 and lysophosphatidic acid receptor LPAR1, the N-terminal helix serves as the cap to seal the ligand binding pocket to prevent nonspecific binding[30,31]. In many cases, the "cap" or "lid" can also act as additional docking site to strengthen ligand-receptor interaction, and to provide additional specificity for ligand recognition. For instance, in LPAR1, the phosphate head group of antagonist ONO-9780307 forms a network of hydrogen bond interaction with $Y34^{N-term}$, $K39^{N-term}$, and $H40^{N-term}$ of the N-terminal helix (Supplementary Fig. 9b)[31]. Similarly, in S1PR1, the $K34^{N-term}$ and $Y29^{N-term}$ of the N-terminal helix interact with the phosphate head of ML056 (Supplementary Fig. 9c)[30]. In CB1, although there is no polar interaction between the agonist AM11542 and the N-terminal helix, the head of the tricyclic tetrahydocannabinol ring inserted into a hydrophobic cavity formed by the N-terminal helix, TM2, TM3, and ECL2[27], makes extensive hydrophobic interaction with $I105^{N-term}$, $F108^{N-term}$, $M109^{N-term}$, $F177^{2.64}$, $F189^{3.25}$, and $F268^{ECL2}$, which firmly lock the upper part of ligand in the pocket (Supplementary Fig. 9d). Aligning BLT1 with those lipid receptors, the carboxyl head group of LTB4 buries much deeper into the pocket and are mainly maintained by the hydrophobic interaction from the barrel of the TM bundle, but not the lid of the receptor (no cap). One major issue of BLT1-targeting anti-leukotriene drug development is the poor efficacy and a lack of specificity, which obstruct the final success in clinical trials. On the contrary, cysteinyl leukotriene receptors have a lid on the top of the ligand

binding pocket (Supplementary Fig. 9a), which provide additional docking site for affinity and specificity in ligand design[29], and this may partially explain why anti-leukotriene drug development succeeds in CysLT, but not BLT[3,32]. We then asked whether the wide-open upper pocket is common in BLT1 and BLT2. Since there is no BLT2 structure available, we use Alpha Fold model[33] of BLT2 (AlphaFold code: AF-Q9NPC1-F1) for comparison. The Alpha Fold BLT2 aligns well with BLT1 (RMSD of 1.168 Å over 199 Cα, Supplementary Fig. 9e), the upper ligand-binding pocket is very similar to BLT1, suggesting that the ligand-binding pocket of BLT2 is also widely open.

## Discussion

Taken together, in this study we reveal the key determinants for LTB4 binding to BLT1, in which water molecules mediated a dynamic hydrogen bond network to connect the receptor with LTB4. We find that the middle C5 hydroxyl group of LTB4 is at the center of the hydrogen-bond network. As discussed earlier, a key difference between LTB4 and 12-HHT is the lack of the middle C5 hydroxyl group in 12-HHT, and this explains why 12-HHT has a much lower affinity than LTB4 in BLT1. We also unveil the activation mechanism for BLT1, together with the discovery of a PI binding site, our study boosts our understanding of LTB4 signaling and provides a rational basis for the design of novel anti-leukotriene drugs.

## Methods

**Constructs**. The codon-optimized human BLT1R gene with a HA-signal peptide sequence at its N-terminus and a LgBiT[11] fusion to the C-terminus, followed by a Tobacco etch virus (TEV) cutting site and 2 x maltose-binding protein (MBP) was cloned in pFastBac1 baculovirus expression vector. Four thermostabilizing mutations $L106^{3.41}W$, $A196^{5.53}I$, $C287^{7.55}F$, and S310A in BLT1 adopted from the MK-D-046-bound BLT1 crystal paper[9] was introduced the receptor. The C-terminus HiBiT fusion of human Gβ1 was cloned into pFastBac plasmid as the VIP1R paper[11]. The human dominant-negative Gα$_{i1}$ (S47N, G203A, A326S and E245A)[12,34], wild-type human Gβ1, wild-type human Gγ2 were cloned into pFastBac plasmid. The scFv16 encoding the single-chain variable fragment of mAb16[13] as described before[35].

**Expression and purification of BLT1/$G_i$ complex**. For expression, baculovirus encoding the BLT1, Gα$_{i1}$, Gβ1, Gγ2, and scFv16 protein were co-infected into the Spodoptera frugiperda (Sf9) cells ($2 \times 10^6$ cells per ml) at a ratio of 1:100 (virus volume vs cells volume) and cells were harvested 48 h postinfection. Cell pellets were resuspended in 20 mM Hepes buffer, 150 mM NaCl, 10 mM $MgCl_2$, 20 mM KCl, 5 mM $CaCl_2$, pH 7.5, with 0.5 mU/ml apyrase and homogenized by douncing ~30 times. To keep the complex stable, ligands was added at final concentration of 300 nM all through the purification. After 1 h incubation of the lysis at room temperature, 0.5% (w/v) lauryl maltose neopentylglycol (LMNG, Anatrace), 0.1% (w/v) cholesteryl hemisuccinate TRIS salt (CHS) were added to solubilize the membrane at 4 °C for 2 h. Then the lysis was ultracentrifuged at 65,000 g at 4 °C for 40 min. The supernatant was incubated with amylose column for 2 h then washed with a buffer of 25 mM Hepes, pH 7.5, 150 mM NaCl and 0.01% LMNG/0.002% CHS, and eluted with the same buffer plus 10 mM maltose. The elute was concentrated and cut with home-made TEV overnight at 4 °C, then separated on a Superdex 200 Increase 10/300 GL (GE health science) gel infiltration column with a buffer of 25 mM Hepes, pH 7.5, 150 mM NaCl and 0.00075% (w/v) LMNG, 0.00025% glyco-diosgenin (GDN), 0.0002% (w/v) CHS (Anatrace). The BLT1/$G_i$ complex corresponding peak was concentrated at about 10 mg/ml and snap frozen for later cryo-EM grid preparation.

**Grid preparation and cryo-EM data collection**. A 3 μl BLT1/$G_i$ complex sample (~10 mg/ml) was applied to a glow-charged quantifoil R1.2/1.3 Cu holey carbon grids (Quantifoil GmbH). The grids were vitrified in liquid ethane on a Vitrobot Mark IV (Thermo Fisher Scientific) instrument at the setting of blot force of 10, blot time of 5 sec, the humidity of 100%, the temperature of 4 °C. Grids were first screened on a FEI 200 kV Arctica transmission electron microscope (TEM) and then grids with evenly distributed particles in thin ice were transferred to a FEI 300 kV Titan Krios TEM equipped with a Gatan Quantum energy filter. Images were taken by a Gatan K2 direct electron detector at the magnitude of 64,000, super-resolution counting model at pixel size of 0.55 Å, the energy filter slit was set to 20 eV. Each image was dose-fractioned in 40 frames using a total exposure time of 7.3 sec at a dose rate of 1.5 e/Å²/s (total dose 60 e/Å²). All image stacks were

collected by the FEI EPU program, nominal defocus value varied from −1.2 to −2.2 μm.

**Data processing**. We use a similar pipeline to process data as descripted before[35]. A total of 3158 raw movies (0.55 Å) were binned (1.1 Å) and motion-corrected using MotionCor2[36], followed by CTF estimation by CTFFIND 4.1[37]. Particles (1.5 million) were picked by crYOLO[38] and extracted by RELION (version 3.1)[39,40] and subjected to reference-free 2D classification in RELION. 12 classes (total about 750,000 particle), which of well-defined features were passed to the next round for initial model generation and 3D classification. The initial model was generated by cryoSPARC[41] ab initio. The model was used as reference in RELION 3D classification (5 classes). One class (of total 500,000) showed clear secondary structure features was selected for a 3D refinement in RELION, followed by a Baysian polishing[42], then a 3D refinement and a CTF refinement in RELION. The refined particles were subjected to a second-round 3D classification (4 classed) with fine angular sampling to yield a class of about 410,000 particles for final refinement by the cryoSPARC Nonuniform Refinement, which generated a map of 2.91 Å, based on the gold standard Fourier Shell Correlation (FSC) = 0.143 criterion. Local resolution estimations were performed using an implemented program in cryoSPARC.

**Model building**. The crystal structures of MK-D-046-bound human BLT1 (PDB 7k15)[9] and the $G_i$ protein complex from the DRD2 (PDB 6vms)[23] were used as initial models for model rebuilding and refinement against the electron microscopy map. All models were docked into the electron microscopy density map using UCSF Chimera[43] then subjected to iterative manual adjustment in Coot[44], followed by a rosetta cryoEM refinement[45] at relax model and Phenix real space refinement[46]. The model statistics were validated using MolProbity[47]. Structural Figures were prepared in UCSF Chimera, ChimeraX[48], and PyMOL (https://pymol.org/2/).

**Structure and sequence comparison**. The calculation of the pocket volume was done by the CASTp 3.0 sever[49]. Sequence alignment by the Clustal Omega sever[50] and the representation of sequence alignment was generated using the ESPript website[51] (http://espript.ibcp.fr). The generic residue numbering of GPCR is based on the GPCRdb4 (https://gpcrdb.org/)[52].

**The SRE-reporter assay**. The SRE reporter assays (Promega) were performed by the Promega instruction as described before. Briefly, AD293 cells were split into 24 well plate at a density of 40,000 per well then transfected with 100 ng of SRE-Luc, 10 ng of pcDNA3-BLT1 wild-type or mutations, 10 ng of phRGtkRenilla plasmids (Promega) by X-tremeGENE HP (Roche) at a ratio 3:1 to DNA amount after one day of growth on 37 °C at 5% $CO_2$. 16 h after transfection, cells were induced by LTB4 (300 nM) or vehicle. Cells were harvested and lysed by addition of 1× Passive Lysis Buffer (Promega) 6 h after induction. The luciferase activity was assessed by the Dual-Glo Luciferase system (Promega). Data were plotted as firefly luciferase activity normalized to Renilla luciferase activity in Relative Luciferase Units (RLU).

**Lipid analysis via Mass Spectrometry**. The purified BLT1 sample (78 μg) was mixed with 1365 μL ice-cold solution containing methyl tert-butyl ether/methanol/ 2 M hydrochloric acid solution (200/60/13, v/v/v) and vortexed for 1 min. Then, the solution was added with 250 μL hydrochloric acid (0.1 M) and vortexed for 5 min. After 5 min centrifugation (6000 g), the upper layer of the solution, which contained the extracted lipids was pipetted out and dried under nitrogen flow. The lipid extracts were dissolved in 100 μL solution containing isopropanol/methanol/ 13 M ammonia solution/chloroform (4/2/0.015/1, v/v/v/v) for subsequent LC-MS analysis. LC-MS was conducted on a Shimadzu LC-20AD system (Kyoto, Japan) hyphenated with a QTRAP 4500 mass spectrometer (Sciex, Toronto, Canada). The injection volume was 2 μL. A hydrophilic interaction chromatography column (150 mm × 2.1 mm, 2.7 μm, Sigma-Aldrich, MO, USA) was used for the separation of glycerophospholipids by class. A binary mobile phase system was used with A being 10 mM ammonium acetate aqueous solution and B being acetonitrile. The flow rate was set at 0.2 mL/min and the oven temperature was kept at 32 °C. The mobile phase gradient was as follows: 90% B for 0–2 min, 90–85 % B for 2–5 min, 85–80% B for 5–8 min, 80% for 8–15 min, 80–70% for 15–16 min, 70% for 16–20 min, 70–90% for 20–21 min, and 90% for 23–25 min. The MS parameters were as follows: voltage for electrospray ionization, −4400 V; curtain gas, 30 psi; interface heater temperature, 450 °C; nebulizing gas 1 and gas 2, 30 psi; declustering potential, −100 V; CID energy at −40 eV.

**Molecular dynamics simulation**. The cryo-EM structure of BLT1 (receptor only) was used to initial model in the MD simulation. The five water molecules and LTB4 were kept in the model while CHS and PI were deleted. Using CHARMM-GUI[53,54], the complex was inserted into a bilayer lipid contain POPC (palmitoyl-2-oleoyl-sn-glycero-3-phosphocholine) and cholesterol at ratio of 4:1, the membrane size is 65 × 65 Å with 22.5 Å water and ion 0.15 M KCl in the top and bottom, temperature 303.15 K. The Amber force fields were set to protein FF19SB, lipid LIPID17, water TIP3P and ligand GAFF2. The simulations were performed by

Amber20 package[55]. The system was first energy minimized for solvent and all atoms, heat to 300 K in 300 ps and then equilibrated for 700 ps, followed by three independent production runs of 200 ns with a timestep of 2 fs. During simulations, Particle mesh Ewald algorithm were applied for the calculation of long-range electrostatic interaction and a cutoff of 10 Å were applied for short-range electrostatic interaction and van der Waals interactions. All bonds with hydrogens are constrained by SHAKE algorithm. The system temperature (300 K) and pressure (1 atm) were controlled by Langevin thermostat and Berendsen barostat, respectively. The trajectories were analyzed by CPPTRAJ[56] and visualized in VMD[57]. The video was recorded by VMD.

**Docking studies**. Based on the cryo-EM structure, LTB4 were re-docked to its pocket. After the preparation and minimization of BLT1 with waters, LTB4 were firstly placed in the pocket using the triangle matcher with London docking scoring. Then, refinement was employed based on a rigid receptor and GBVI/WSA docking scoring. The top-scored pose was shown in Supplementary Fig. 5c.

**Reporting summary**. Further information on research design is available in the Nature Research Reporting Summary linked to this article.

## Data availability

All data produced or analyzed in this study are included in the main text or the supplementary materials. A reporting summary for this article is available as a Supplementary Information file. The cryo-EM density maps and atomic coordinates have been deposited in the Electron Microscopy Data Bank (EMDB) and Protein Data Bank (PDB) under accession numbers EMD-32018 and 7VKT [https://www.rcsb.org/structure/7VKT] for the LTB4-bound BLT1/$G_i$ complex. All other published PDB codes cited in this paper are 7K15, 5×33, 5XRA, 6AK3, 6RZ6, 3V2Y, 4Z34 and 6VMS. Source data are provided with this paper.

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

## Acknowledgements

We thank Anqi Zhang, Youpeng Qu and Changyou Guo of the Cryo-EM facility of Harbin Institute of Technology for sample screening and data collection. This work was supported by the Startup Funds of HIT Center for Life Sciences; the National Natural Science Foundation of China (32070048 to Y.H.); Ministry of Science and Technology of China (2018YFA0507002 to H. E. X.), Shanghai Municipal Science and Technology Major Project (2019SHZDZX02 to H. E. X.), and CAS Strategic Priority Research Program (XDB08020303 to H. E. X.); National Science & Technology Major Project "Key New Drug Creation and Manufacturing Program" of China (2018ZX09711002 to H.J.); Science and Technology Commission of Shanghai Municipal (20431900100 to H.J.); Jack Ma Foundation (2020-CMKYGG-05 to H.J.). X.Y and Z.J. acknowledge the financial support by the National Key R&D Program of China (2018YFA0800903). We thank Dr. Zhiwei Huang of the HIT Center for Life Sciences for suggestion and support of this project. We thank Dr. David A. Case (Rutgers University) and the Amber Organization for the support of Amber MD package.

## Author contributions

N.W. made the expression constructs, purified the proteins, prepared and screened the grids, made the mutations, performed functional assays and analyzed data. X.H. analyzed the data and performed docking and MD simulation studies. J.Z. and Y.X. performed MS analysis. H.J., X.C., Y.X., and H.E.X. analyzed data and supervise the experiments with Y.H. Y.H. conceived the project, designed the experiments, analyzed data, collected data, solved the structures, performed MD, wrote the manuscript, and supervised the project. All authors contributed to data interpretation and preparation of the manuscript.

## Competing interests

The authors declare no competing interests.
