## [Peer Review File · Nature Communications]

Reviewer #1 (Remarks to the Author):

The manuscript described the activate BLT1 complex the intrinsic ligand LTB4 structure with Gi1-protein supported with scFv16, and PtdIns4P at intracellular side of BLT1 by cryo-EM.

It is regretfully to reach my conclusion that the manuscript is reject or major revision needed, due to neither sound or confident with concrete experimental evidence in the many parts of the description. Some corrections are pointed separately.

First of all, I was surprised in the unusual interaction both intrinsic bound ligands LTB4 and PtdIns4P whose negative charged group shows no clear interactions with any counter charged groups of BLT1 (main text lines 93-95) exceptionally as shown intrinsic ligands such as serotonin (ref. 15) or PGE2 (PDB code 6AK3, 7D7M and 7CX2).

Additionally, the presented chemical structure is incorrect in the manuscript. BLT4 in Fig. 2a, the stick model of C4 and C12 seems to be the sp³ hydroxyl-substituted carbon atoms due to the correct assigned to PDB chemical 3 letters code "LBT" as the residue 401 in the attached PDB validation report, whereas in Fig. 2b, the chemical structure of BLT4 was drawn both carbons as carbonyls. The structural evidence were not sufficiently confident including these polar interactions as described in main text as follows.

"Interestingly, the greatest decrease of activity is seen on the polar mutant R156A, which almost completely killed the receptor. The R156 forms a key hydrogen bond with the carbonyl sulfonamide group of MK-D-046 in the crystal structure (Fig. 3b)." (main text lines 105-108)

"We did not see a direct polar interaction between R156 and LTB4, however, it is in the proximity of the C12 carbonyl (hydroxy) group (Fig 2a), and current resolution we cannot rule out a water molecule may mediate this interaction." (main text lines 108-110).

In the manuscript, the density maps were well fitted in their 7 transmembrane helices including their side-chains in Supplementary Fig. 4, whereas LTB4 and PtdIns4P density were not shown to be fully fitted their functional group hydroxyl or phosphate. These functional group, especially for phosphate group in PtdIns4P, should have higher density as shown the cited serotonin receptor complex structure (ref 15). In protein structural determination, one of most difficult and ambiguous part(s) is ligand identification, in particular, unexpectedly found one like PtdIns4P as described in the manuscript due to the relative low affinity and not fully occupancy in comparison with synthetic designed chemical ligands like BIIL260 (ref. 7) or MK-D-046 (ref. 8) as BLT1 inverse agonist or antagonist for drug candidates to achieve the higher target selectivity and affinity among GPCRs. However, in most structure determined complex structures, intrinsic ligands showed well defined structures with several intimate strict interactions with their charged or polar groups to the counter charged or polar groups in the bound GPCRs as shown serotonin, PGE2 or histamine with much lower affinities. Thus most GPCR structures were obtained as complexes with those drug candidate ligands at first.

"A comparison of the two PtdIns4P binding sites shows significant difference Nevertheless, the adding of PtdIns4P to HT1A/Gi complex also positively regulates receptor activation, consistent with our observation in BLT1." (main text line 133-141)

In my opinion, the authors should specify and confirm to show experimentally the interactions between BLT1 and PdtIns4P structurally and chemically as well as functionally in the revised version of manuscript, because I could not find any reports on the interaction between BLT1 and PdtIns4P so far.

The section of "Methods" is almost completely the same except for some specific experimental conditions as the previous published paper which was the authors of the manuscript are co-authors, I recommend to rewrite according to following the experimental process in the

manuscript.

Recommendation to Correct

line 183-184 ... a water or an anion molecule ... >> ... a water or a cation... or ... a water or a sodium cation ...

Fig. 2b

The chemical structure of LTB4 was incorrectly presented, that is, 4(S),12(R)-hydroxyl groups of LTB4 were shown incorrect as carbonyls in Fig. 2b and main text as shown above.

General suggestion: The whole text may be English re-writing is recommended to improve the English by English native Expert. Unfortunately I cannot revise to improve English expression of the manuscript, while I am not native speaker.

Reviewer #2 (Remarks to the Author):

This manuscript by Wang & He reports on the cryoEM structure of the high-affinity LTB4 BLT1 G Protein-Coupled Receptor at a resolution of $\sim 3\text{\AA}$. In addition to the description of the complex, they also identified a lipid cofactor which has positive allosteric properties in the activation of the receptor. This work represents an important contribution to the field as it describes for the first time an active conformation for this receptor in complement to the two inactive structures already described in the literature. On a technical point of view this work seems well done, including state-of-the-art subterfuges like NanoBiT technology to get very nice objects to look at. That would have been awesome to have a comparative study with BLT2 receptor, but I fully understand that with BLT2 the task could be much more complicated. However, beyond purely x,y,z description, I think a deeper analysis in term of structural determinants for LTB4 and Gi selectivity would be beneficial for this paper. By the way, overall, the data is very convincing but the manuscript would gain in quality after reworking the analysis and the writing.

Additional comments:

1. As an editorial axis, the authors introduce BLT1 on a comparative basis with BLT2 receptor which is an analog receptor that binds eicosanoid, including LTB4, but not only. I think this introduction suffers from several shortcuts. For instance, "BLT1 has a much higher affinity to LTB4 than BLT2". Well, this is true, but this is also misleading as BLT1 is activated by the eicosanoid LTB4 only, not BLT2. In addition, several more or less recent studies highlight that the endogenous agonist for BLT2 would not be LTB4 but the heptadecanoid 12-HHT for which it displays an affinity comparable to BLT1 for LTB4. Indeed, the K_d of BLT2 for LTB4 is pretty low which probably means that LTB4 is probably not an endogenous agonist for this receptor.

By the way, the choice of introducing BLT1 in a comparison fashion with BLT2 is logic and interesting, but, if so, this should be discussed further in the manuscript, especially in the § entitled "The upper open pocket of BLT1".

2. I think the authors should be more careful regarding the molecular interactions between the agonist and the amino acids lining the pocket. In such a wide pocket, where resides the ligand selectivity? Just on hydrophobic interactions? Based on their description, we can't put that data in perspectives with for instance competition assays that have been performed in addition to LTB4 with structurally related eicosanoids (e.g. Sabirsh et al. *Biochemistry* 2006, 45, 5733-5744). For instance, it's intriguing that no interactions are observed between the two hydroxyl groups and/or the carboxylic function of LTB4 with BLT1 knowing that using other eicosanoids where one of the hydroxyl group is missing for instance considerably modifies the affinity for this receptor. Directed single mutagenesis associated with luciferase system assay are interesting, but, overall, unfortunately this is still difficult to learn from it about ligand selectivity.

On a technical point of view : looking at Fig. S4, the LTB4 ligand fits well in the e density, but at such a resolution, this does not preclude that the ligand could be slightly shifted which could cause

it not to interact with fundamental amino acids. In addition, LTB4 does not have a sufficiently characteristic structure (as this is the case with a haem for example) for its identification to be unambiguous at this resolution (how to distinguish from a fatty acid or a detergent molecule?) And it also depends on the occupancy rate of the ligand if it is less than 1.0 (?). Finally, in EM, electrons are deflected by charges, this makes identification of carboxylates more difficult at low resolution (at high resolution there is in principle no problems).

To summarize all of this, I think these points should be discussed somewhere in the manuscript.

In addition the resolution should be indicated at least in the introduction if not in the abstract.

3. The complex assembling combined the use of state-of-the-art technologies like NanoBiT and the introduction of mutations to facilitate expression and stabilization of the assembly for proper cryoEM structural investigations. The authors mention that these four mutations have been used in the BLT1 construct to get its crystal structure associated with an antagonist, so in an inactive state. So, at a first sight, that sounds paradoxical, so maybe the authors should be more explanatory. Actually, in the study of MK-D-046/BLT1 by Michaelian et al, they introduced 5 mutations, and more precisely five thermostabilizing mutations (and I think the qualifier "thermostabilizing" should be also mentioned in the present manuscript). All five mutations combined decreased the efficacy and potency of the agonist LTB4 but had almost no incidence on the potency and inhibition efficacy of the antagonist MK-D-046. But, as underlined in Michaelian et al, this reduction is mainly due to a fifth mutation (S116Y) which is not present here. So maybe this should be more explicitly written to help the reader to understand the introduction of these thermostabilizing mutations that seem to essentially improve the expression level. They should also comment why in Fig. S1b these 4 mutations, without the presence of LgBiT, but also in a lesser extent in the presence of the nanobit, do boost Gi signaling as indicated in the SRE reporter assay compared to the wt. The acronyms SRE-luc and RLU should be clearly explained in the caption without the need for the reader to have a look in the methods section.

A typo: the fourth mutation is indicated to be S301A (line 60) and in Michaelian et al and in the Methods of the manuscript this is S310A (overall there are some typos occasionally in the manuscript, please have a look carefully).

4. The impact of allosteric modulators on GPCR activities, and in particular of lipids, is a fundamental aspect and represents a certain added value in this study. Again, above x,y,z considerations, I think that part of the study would deserve a deeper analysis based on the abundant literature available. Their results should be put in a more perspective way with for instance such papers like: Yen, H. Y. et al. PtdIns(4,5)P2 stabilizes active states of GPCRs and enhances selectivity of G-protein coupling. *Nature* 559, 423–427 (2018) or Damian M et al. Allosteric modulation of ghrelin receptor signaling by lipids. *Nat Commun.* 12, 3938 (2021). In Damian et al, PIP2 shifts the conformational equilibrium of GHSR away from its inactive state, favoring GPCR activation, including its basal activity. Concerning BLT1, I was wondering if PtdIns4P has also an impact on the basal activity of the receptor? Would it be possible to complete data exposed in Fig. 2G,h?

NB: I think the methods is not described enough for this assay in the absence or presence of PtdIns4P. This should be a little bit more described in the methods section.

5. There is a large part dedicated to the comparison of the orthosteric pocket between the inactive and active conformations and a very short part concerning the rest of the receptor. So, the ligand-induced BLT1 rearrangement leading to Gi coupling remains unclear for me. Nat Comm authorized up to 10 items, so there is plenty of space for additional illustrations if needed.

By the way, interestingly, the authors identified some distinct interaction between BLT1 and Gi compared to the literature. Compared to GPCR-Gs complexes for instance, could be formulated some clues concerning the selectivity of BLT1 for Gi? For example, several studies suggest that Gi-

coupled receptors require a smaller outward movement of TM6 compared to Gs-coupled receptors.

6. About ligand or G selectivity, I think one or two sentences on kinetics of interactions -instead to base all the analysis on equilibrium properties- would be beneficial to the manuscript as this turn to be a fundamental criteria in the activation of the receptor and/or the coupling to a cytosolic partner.

Reviewer #3 (Remarks to the Author):

The manuscript presents the first active state structure of the leukotriene receptor BLT1 in complex with the G protein Gi. The structure shows how the native ligand LTB4 binds to the receptor and how the activation is similar to other Class A GPCRs. Detailed descriptions of the orthosteric binding site and G protein coupling site are given, and mutagenesis data suggests the relative importance of the residues for binding. In addition, two sorts of different densities in the transmembrane region have been assigned to either cholesterol or the phosphatidylinositol PI4.

The manuscript is well presented and the figures are clear. There are a few points that need addressing by the authors.

1. Line 75. How do you know the densities represent cholesterol and not either CHS or GDN, both present in the receptor purification? Please provide mass spectrometry data identifying the presence of only cholesterol in the purified protein. In the absence of this, they should be modelled as CHS unless there are compelling reasons not to do so. These should be described clearly in the methods section and referred to in the main text.
2. Line 88. 'Flip-down L shape' is unhelpful. It is probably better just to say that LTB4 binds in an extended conformation with the carboxylate in the aqueous environment at the entrance to the orthosteric binding pocket.
3. Lines 89-90. This list does not correspond exactly to that in Fig 2b
4. Line 106. 'Kills' is unscientific; replace here and elsewhere with an accurate term.
5. Line 120. How do you know that the lipid is PI4P Ideally mass spectrometry data should be used to positively identify the lipid. In the absence of such data, it should be referred to here and elsewhere as 'putative PI4P'. Please state in the text why it cannot be any other phosphatidyl inositol; for example, is it impossible for any other phosphate groups to be present because of steric clashes with the receptor due to the shape of the pocket it binds in?
6. Lines 118 and 119 (and throughout the manuscript); the use of possessives (proteins', GPCRs') is grammatically correct but may be confusing for those with a more tenuous grasp of English. Reformulate.
7. Line 138. 'Den', is unscientific; reformulate.
8. Line 184, 'inion' should be 'anion'
9. Line 223. It is untrue that 'most class A GPCRs' have a cap or lid over the orthosteric binding site in the active state. It might be true for lipid-binding receptors, but 'some' is a better description for all of Class A.
10. Line 314. No chemical is given for the compound at 0.0002% in the buffer; is this CHS?
11. Line 344; RELOIN is spelt wrong
12. Line 411; the residues probably do not block the 'entry' of water so much as prevent the presence of water sterically.
13. Line 443. There is no description of panel c.
14. Line 449. Choose better cut-off values so that the local resolution analysis on the model is not all blue. Ideally only about 30-50% should be blue so it gives a clearer picture of the resolution throughout the density.
15. Line 456. The cut-off at the extracellular region for the density determination of the orthosteric binding pocket needs to be chosen more carefully. The spheres of volume suggests that regions extracellular to the OBP are being included in the calculation.

16. Line 474. Please use spheres to represent the ligands so they can be seen more easily.
17. Fig 2h. Give a statistical analysis for whether the differences observed between vehicle and PI4P are significant or not. How do you know that in the vehicle there is no PI4P in the receptor preparation? You might only be observing a small increase in activity because there is already PI4P in most of the receptor molecules. Discuss.
18. For all bar graphs, give a statistical analysis for the significance of the differences observed.

REVIEWER COMMENTS

Reviewer #1 (Remarks to the Author):

The manuscript described the activate BLT1 complex the intrinsic ligand LTB4 structure with Gi1-protein supported with scFv16, and PtdIns4P at intracellular side of BLT1 by cryo-EM.

It is regretfully to reach my conclusion that the manuscript is reject or major revision needed, due to neither sound or confident with concrete experimental evidence in the many parts of the description. Some corrections are pointed separately.

We have made major changes on our manuscript, and most importantly, we did in-depth data analysis, including running MD simulation, docking, and additional experiments, including mass spectrometry and site directed mutagenesis, to validate our data.

First of all, I was surprised in the unusual interaction both intrinsic bound ligands LTB4 and PtdIns4P whose negative charged group shows no clear interactions with any counter charged groups of BLT1 (main text lines 93-95) exceptionally as shown intrinsic ligands such as serotonin (ref. 15) or PGE2 (PDB code 6AK3, 7D7M and 7CX2).

First, we identified a cluster of water molecules density via a careful examination of our cryo-EM data. These water molecules form a network of hydrogen bond interactions that connects the key polar residues (i.e. R156 and H94) to LTB4 in the pocket (Fig. 2a). In addition, we identified another key residue R267 play key role in the network, and mutation of R267A cause a total loss of receptor activity (Fig. 2a, 2d). Furthermore, we use molecule dynamic simulations to examine the stability of LTB4 in the ligand binding pocket, and the data shows that LTB4 is very stable in the ligand binding pocket (Fig. 2c, S5b, and Supplementary video 1), the conformations of LTB4 in the simulation are very stable and similar to the original conformation of LTB4 we observed in our cryo-EM structure. Importantly, we observed a very dynamic water hydrogen network that connects LTB to receptor in the simulation. We also use glide docking to analyze LTB binding, and the data totally agree with our structural observation. Regarding about PtdIns4P, we used mass spectrometry to find out the corresponding molecules are actually PI 18:0/16:1 or PI 18:0/18:1, not the PtdIns4P. We have corrected the mistake. We thank you very much for your comments, we feel that those comments greatly improve the accuracy and quality of our paper.

Additionally, the presented chemical structure is incorrect in the manuscript. BLT4 in Fig. 2a, the stick model of C4 and C12 seems to be the sp³ hydroxyl-substituted carbon atoms due to the correct assigned to PDB chemical 3 letters code "LBT" as the residue 401 in the attached PDB validation report, whereas in Fig. 2b, the chemical structure of BLT4 was drawn both carbons as carbonyls. The structural evidence were not sufficiently confident including these polar interactions as described in main text as follows.

We thank you very much for pointing out this. We have corrected this in the new Fig. 2b.

"Interestingly, the greatest decrease of activity is seen on the polar mutant R156A, which almost completely killed the receptor. The R156 forms a key hydrogen bond with the carbonyl sulfonamide group of MK-D-046 in the crystal structure (Fig. 3b)." (main text lines 105-108)
"We did not see a direct polar interaction between R156 and LTB₄, however, it is in the proximity of the C12 carbonyl (hydroxy) group (Fig 2a), and current resolution we cannot rule out a water molecule may mediate this interaction." (main text lines 108-110).

As mentioned early, we have identified a cluster of water molecules that mediated the polar interaction between receptor and LTB₄ via a careful examination of our density map. Importantly, we also identified another key residue, R267, mediated the interaction (Fig. 2a, 2d). Those observations are validated by MD simulations and docking, and fully supported by our mutagenesis studies (Fig. 2c, s5b-c, 2d, video 1).

In the manuscript, the density maps were well fitted in their 7 transmembrane helices including their side-chains in Supplementary Fig. 4, whereas LTB₄ and PtdIns4P density were not shown to be fully fitted their functional group hydroxyl or phosphate. These functional group, especially for phosphate group in PtdIns4P, should have higher density as shown the cited serotonin receptor complex structure (ref 15). In protein structural determination, one of most difficult and ambiguous part(s) is ligand identification, in particular, unexpectedly found one like PtdIns4P as described in the manuscript due to the relative low affinity and not fully occupancy in comparison with synthetic designed chemical ligands like BIL260 (ref. 7) or MK-D-046 (ref. 8) as BLT1 inverse agonist or antagonist for drug candidates to achieve the higher target selectivity and affinity among GPCRs. However, in most structure determined complex structures, intrinsic ligands showed well defined structures with several intimate strict interactions with their charged or polar groups to the counter charged or polar groups in the bound GPCRs as shown serotonin, PGE₂ or histamine with much lower affinities. Thus most GPCR structures were obtained as complexes with those drug candidate ligands at first.

We admit that we built the PtdIns4P ambiguously without support of strong density and experimental evidence. We have corrected this through a thorough examination of the lipid components of our complex via mass spectrometry. We have obtained clear and strong experimental evidence that the binding lipid is PI instead of PtdIns4P (Fig. 2f, s6a-b). We thank you again for your comments.

"A comparison of the two PtdIns4P binding sites shows significant difference Nevertheless, the adding of PtdIns4P to HT1A/Gi complex also positively regulates receptor activation, consistent with our observation in BLT1." (main text line 133-141)

In my opinion, the authors should specify and confirm to show experimentally the interactions between BLT1 and PdtIns4P structurally and chemically as well as functionally in the revised

version of manuscript, because I could not find any reports on the interaction between BLT1 and PdtIns4P so far.

We made a mistake to model PtdIns4P into the map. We have corrected this via modeling PI 18:0/16:1 into the map (Fig. 2e). Mutations of the allosteric binding pocket, R129A, M156K and D114A, greatly decrease the receptor activity, suggesting a function of the allosteric binding site (Fig. s5d). And to our knowledge this is also the first-time report that PI (18:0/16:1 or PI 18:0/18:1) allosterically binds to GPCR. However, both PI 18:0/16:1 and PI 18:0/18:1 are hardly to obtain, the full function of PI 18:0/16:1 and PI 18:0/18:1 to BLT1 need a thorough investigation in the future.

The section of "Methods" is almost completely the same except for some specific experimental conditions as the previous published paper which was the authors of the manuscript are co-authors, I recommend to rewrite according to following the experimental process in the manuscript.

We have rewrote the methods part.

Recommendation to Correct

line 183-184 ... a water or an inion molecule ... >> ... a water or a cation... or ... a water or a sodium cation ...

Corrected.

Fig. 2b

The chemical structure of LTB4 was incorrectly presented, that is, 4(S),12(R)-hydroxyl groups of LTB4 were shown incorrect as carbonyls in Fig. 2b and main text as shown above.

Corrected, thank you.

General suggestion: The whole text may be English re-writing is recommended to improve the English by English native Expert. Unfortunately I cannot revise to improve English expression of the manuscript, while I am not native speaker.

We have rewrote most part of the main text, thank you.

Reviewer #2 (Remarks to the Author):

This manuscript by Wang & He reports on the cryoEM structure of the high-affinity LTB4 BLT1 G Protein-Coupled Receptor at a resolution of ~3Å. In addition to the description of the complex, they also identified a lipid cofactor which has positive allosteric properties in the activation of the receptor. This work represents an important contribution to the field as it describes for the

first time an active conformation for this receptor in complement to the two inactive structures already described in the literature. On a technical point of view this work seems well done, including state-of-the-art subterfuges like NanoBiT technology to get very nice objects to look at. That would have been awesome to have a comparative study with BLT2 receptor, but I fully understand that with BLT2 the task could be much more complicated. However, beyond purely x,y,z description, I think a deeper analysis in term of structural determinants for LTB4 and Gi selectivity would be beneficial for this paper. By the way, overall, the data is very convincing but the manuscript would gain in quality after reworking the analysis and the writing.

We thank you for your positive comments on our work. We have done further in-depth analysis of our data and additional experiments to improve the accuracy and quality of our paper.

Additional comments:

1. As an editorial axis, the authors introduce BLT1 on a comparative basis with BLT2 receptor which is an analog receptor that binds eicosanoid, including LTB4, but not only. I think this introduction suffers from several shortcuts. For instance, “BLT1 has a much higher affinity to LTB4 than BLT2”. Well, this is true, but this is also misleading as BLT1 is activated by the eicosanoid LTB4 only, not BLT2. In addition, several more or less recent studies highlight that the endogenous agonist for BLT2 would not be LTB4 but the heptadecanoid 12-HHT for which it displays an affinity comparable to BLT1 for LTB4. Indeed, the Kd of BLT2 for LTB4 is pretty low which probably means that LTB4 is probably not an endogenous agonist for this receptor.

We have corrected this in the introduction part by deleting the sentence of “the proinflammatory effect of LTB4 is generally believed to be mainly mediated by BLT1”, and adding more background of BLT2. Thank you.

By the way, the choice of introducing BLT1 in a comparison fashion with BLT2 is logic and interesting, but, if so, this should be discussed further in the manuscript, especially in the § entitled “The upper open pocket of BLT1”.

We did a comparison of BLT1 with BLT2 (alpha fold model), the comparison suggested that BLT2 also has an open upper ligand binding pocket (Fig. S9b).

2. I think the authors should be more careful regarding the molecular interactions between the agonist and the amino acids lining the pocket. In such a wide pocket, where resides the ligand selectivity? Just on hydrophobic interactions? Based on their description, we can't put that data in perspectives with for instance competition assays that have been performed in addition to LTB4 with structurally related eicosanoids (e.g. Sabirsh et al. Biochemistry 2006, 45, 5733-5744). For instance, it's intriguing that no interactions are observed between the two hydroxyl groups and/or the carboxylic function of LTB4 with BLT1 knowing that using other eicosanoids where

one of the hydroxyl group is missing for instance considerably modifies the affinity for this receptor. Directed single mutagenesis associated with luciferase system assay are interesting, but, overall, unfortunately this is still difficult to learn from it about ligand selectivity.

We have carefully examined the density of the cryo-EM map and identified a cluster of 5 water molecules right above LTB4 in the upper ligand binding pocket. Particularly, water molecule 1 (W1) play a pivot role in connecting both R156 and R267 to the C5 hydroxyl group of LTB4 (Fig. 2a). We further use MD simulation to validate our ligand position. The simulation data shows that LTB4 is very stable in the ligand binding pocket, and there is a dynamic water hydrogen network to connect the key polar residues to LTB (Fig. 2c, supplementary video 1 and supplementary Fig. S5b). And the data is also supported by docking study (Fig. S5c) and new mutation study of R267A (Fig. 2d). Thank you for your comment, this greatly improve the accuracy and quality of our paper.

On a technical point of view : looking at Fig. S4, the LTB4 ligand fits well in the e density, but at such a resolution, this does not preclude that the ligand could be slightly shifted which could cause it not to interact with fundamental amino acids. In addition, LTB4 does not have a sufficiently characteristic structure (as this is the case with a haem for example) for its identification to be unambiguous at this resolution (how to distinguish from a fatty acid or a detergent molecule?) And it also depends on the occupancy rate of the ligand if it is less than 1.0 (?). Finally, in EM, electrons are deflected by charges, this makes identification of carboxylates more difficult at low resolution (at high resolution there is in principle no problems).

To summarize all of this, I think these points should be discussed somewhere in the manuscript.

Thank you for your suggestion. The issue has been solved by the identification of additional water molecules that bridge the polar interaction between LTB4 and receptor and validate by MD, docking and mutation studies as mentioned above.

In addition the resolution should be indicated at least in the introduction if not in the abstract.

Done.

3. The complex assembling combined the use of state-of-the-art technologies like NanoBiT and the introduction of mutations to facilitate expression and stabilization of the assembly for proper cryoEM structural investigations. The authors mention that these four mutations have been used in the BLT1 construct to get its crystal structure associated with an antagonist, so in an inactive state. So, at a first sight, that sounds paradoxical, so maybe the authors should be more explanatory. Actually, in the study of MK-D-046/BLT1 by Michaelian et al, they introduced

5 mutations, and more precisely five thermostabilizing mutations (and I think the qualifier “thermostabilizing” should be also mentioned in the present manuscript). All five mutations combined decreased the efficacy and potency of the agonist LTB₄ but had almost no incidence on the potency and inhibition efficacy of the antagonist MK-D-046. But, as underlined in Michaelian et al, this reduction is mainly due to a fifth mutation (S116Y) which is not present here. So maybe this should be more explicitly written to help the reader to understand the introduction of these thermostabilizing mutations that seem to essentially improve the expression level. They should also comment why in Fig. S1b these 4 mutations, without the presence of LgBiT, but also in a lesser extent in the presence of the nanobit, do boost G_i signaling as indicated in the SRE reporter assay compared to the wt. The acronyms SRE-luc and RLU should be clearly explained in the caption without the need for the reader to have a look in the methods section.

Thank you for your comments. The idea of using the thermostabilizing mutations is to facilitate receptor folding and expression, as our early attempt of using WT receptor is not very successful. We exclude the S116Y mutation based its property of decreasing receptor potency (Michaelian’s paper). We have used our SRE-reporter assay to reassessing its activity on receptor activity. To our surprise, the 5 mutations construct which includes S116Y did not decrease the transduction of receptor (compare to WT) in the reporter assay (Fig. S1b), although it is less effective as the 4 mutations construct. The difference might due to different methods used in measuring receptor activity (reporter assay vs IP1 production assay).

A typo: the fourth mutation is indicated to be S301A (line 60) and in Michaelian et al and in the Methods of the manuscript this is S310A (overall there are some typos occasionally in the manuscript, please have a look carefully).

Corrected, thank you.

4. The impact of allosteric modulators on GPCR activities, and in particular of lipids, is a fundamental aspect and represents a certain added value in this study. Again, above x,y,z considerations, I think that part of the study would deserve a deeper analysis based on the abundant literature available. Their results should be put in a more perspective way with for instance such papers like: Yen, H. Y. et al. PtdIns(4,5)P₂ stabilizes active states of GPCRs and enhances selectivity of G-protein coupling. *Nature* 559, 423–427 (2018) or Damian M et al. Allosteric modulation of ghrelin receptor signaling by lipids. *Nat Commun.* 12, 3938 (2021). In Damian et al, PIP₂ shifts the conformational equilibrium of GHSR away from its inactive state, favoring GPCR activation, including its basal activity. Concerning BLT1, I was wondering if PtdIns4P has also an impact on the basal activity of the receptor? Would it be possible to complete data exposed in Fig. 2G,h?

NB: I think the methods is not described enough for this assay in the absence or presence of PtdIns4P. This should be a little bit more described in the methods section.

Thank you for your suggestion. The PtdIns4P was built on an ambiguous density without strong physical support or experimental evidence. We have used mass spectrometry to analyze the lipid component of our BLT1/Gi complex; the data clearly show that it is PI (18:0/16:1 or 18:0/18:1), not PtdIns4P (Fig. 2f, Fig. S6a-b). We have corrected this mistake and rebuilt our model with PI 18:0/16:1. Mutation of the key residue in the allosteric binding pocket is harmful for receptor activity (Fig. S6d), suggesting a function for this allosteric binding. To our knowledge, this is also the first time reporting the allosteric binding of PI (18:0/16:1 or 18:0/18:1) to GPCR. However, both PI 18:0/16:1 and PI 18:0/18:1 are hard to get. The function of PI (18:0/16:1 or 18:0/18:1) to receptor activity need a thorough investigation in the future.

5. There is a large part dedicated to the comparison of the orthosteric pocket between the inactive and active conformations and a very short part concerning the rest of the receptor. So, the ligand-induced BLT1 rearrangement leading to Gi coupling remains unclear for me. Nat Comm authorized up to 10 items, so there is plenty of space for additional illustrations if needed.

Thank you for your suggestion. We also compare the region outside of the ligand binding pocket. We found the extracellular side of TM7 has an outward displacement when bound to antagonist MK and the concomitant sinking displacement of ECL3 (Fig S. 8d).

By the way, interestingly, the authors identified some distinct interaction between BLT1 and Gi compared to the literature. Compared to GPCR-Gs complexes for instance, could be formulated some clues concerning the selectivity of BLT1 for Gi? For example, several studies suggest that Gi-coupled receptors require a smaller outward movement of TM6 compared to Gs-coupled receptors.

Thank you for your suggestion. We did a comparison with β 2AR/Gs complex, yes, in the Gs complex, the TM6 show a more pronounced outward displacement than TM6 of the BLT1/Gi complex and the head of α H5 inserted deeper in the Gs complex. We also find there are much polar interactions between the TM5 and the α H5 in the Gs complex than in the Gi complex, which may account for the selectivity of Gi for BLT1 (Fig. S8e).

6. About ligand or G selectivity, I think one or two sentences on kinetics of interactions -instead to base all the analysis on equilibrium properties- would be beneficial to the manuscript as this turn to be a fundamental criteria in the activation of the receptor and/or the coupling to a cytosolic partner.

Based on above comparison, we speculate that the stronger polar interactions between TM5 and the α H5 may account for the higher stability of the Gs complex in many cases. We have discussed this in the main text.

Reviewer #3 (Remarks to the Author):

The manuscript presents the first active state structure of the leukotriene receptor BLT1 in complex with the G protein Gi. The structure shows how the native ligand LTB4 binds to the receptor and how the activation is similar to other Class A GPCRs. Detailed descriptions of the orthosteric binding site and G protein coupling site are given, and mutagenesis data suggests the relative importance of the residues for binding. In addition, two sorts of different densities in the transmembrane region have been assigned to either cholesterol or the phosphatidylinositol PI4.

Thank you for your positive comment on our study.

The manuscript is well presented and the figures are clear. There are a few points that need addressing by the authors.

1. Line 75. How do you know the densities represent cholesterol and not either CHS or GDN, both present in the receptor purification? Please provide mass spectrometry data identifying the presence of only cholesterol in the purified protein. In the absence of this, they should be modelled as CHS unless there are compelling reasons not to do so. These should be described clearly in the methods section and referred to in the main text.

Thank you for your comment and suggestion. We used mass spectrometry to analyze the lipids component of our BLT1/Gi complex. Due to the low ionization efficiency of cholesterol, we did not detect cholesterol above noise level. You are right there are a lot of CHS molecules in the purification buffer. We have corrected this and modeled CHS in our structure.

2. Line 88. 'Flip-down L shape' is unhelpful. It is probably better just to say that LTB4 binds in an extended conformation with the carboxylate in the aqueous environment at the entrance to the orthosteric binding pocket.

Thank you for your suggestion, we have corrected this.

3. Lines 89-90. This list does not correspond exactly to that in Fig 2b

We have corrected this.

4. Line 106. 'Kills' is unscientific; replace here and elsewhere with an accurate term.

We have corrected this.

5. Line 120. How do you know that the lipid is PI4P Ideally mass spectrometry data should be used to positively identify the lipid. In the absence of such data, it should be referred to here and elsewhere as 'putative PI4P'. Please state in the text why it cannot be any other

phosphatidyl inositol; for example, is it impossible for any other phosphate groups to be present because of steric clashes with the receptor due to the shape of the pocket it binds in?

Thank you so much for your suggestion! The PtdIns4P was built on an ambiguous density without strong physical support. To figure out what exactly this density corresponds to, we use mass spectrometry to identify the lipid component of our complex. With clear and strong evidence, we found the density should correspond to PI (18:0/16:1 or 18:0/18:1), not the PtdIns4P (Fig. 2f, Fig. S6a-b). We have corrected this mistake and rebuilt the model with PI 18:0/16:1. We also discussed this in the main text.

6. Lines 118 and 119 (and throughout the manuscript); the use of possessives (proteins', GPCRs') is grammatically correct but may be confusing for those with a more tenuous grasp of English. Reformulate.

We have corrected this.

7. Line 138. 'Den', is unscientific; reformulate.

Since we corrected our mistake of PtdIns4P, we have deleted this sentence.

8. Line 184, 'inion' should be 'anion'

Corrected.

9. Line 223. It is untrue that 'most class A GPCRs' have a cap or lid over the orthosteric binding site in the active state. It might be true for lipid-binding receptors, but 'some' is a better description for all of Class A.

Corrected.

10. Line 314. No chemical is given for the compound at 0.0002% in the buffer; is this CHS?

Yes. We corrected this.

11. Line 344; RELOIN is spelt wrong

Corrected.

12. Line 411; the residues probably do not block the 'entry' of water so much as prevent the presence of water sterically.

We changed to inhibit.

13. Line 443. There is no description of panel c.

Fixed.

14. Line 449. Choose better cut-off values so that the local resolution analysis on the model is not all blue. Ideally only about 30-50% should be blue so it gives a clearer picture of the resolution throughout the density.

Fixed. Thank you.

15. Line 456. The cut-off at the extracellular region for the density determination of the orthosteric binding pocket needs to be chosen more carefully. The spheres of volume suggests that regions extracellular to the OBP are being included in the calculation.

Fixed.

16. Line 474. Please use spheres to represent the ligands so they can be seen more easily.

Fixed.

17. Fig 2h. Give a statistical analysis for whether the differences observed between vehicle and PI4P are significant or not. How do you know that in the vehicle there is no PI4P in the receptor preparation? You might only be observing a small increase in activity because there is already PI4P in most of the receptor molecules. Discuss.

Since we have corrected PI4P to PI, we have deleted this fig.

18. For all bar graphs, give a statistical analysis for the significance of the differences observed.

Done, thank you.

Reviewer #1 (Remarks to the Author):

The major revised version is the many improvements of BLT1 complex structure with Gi with cryoEM in the various features according to the other reviews including my opinion, and the quality reached to meet to publish in a well-known journal like Nature communication with obvious importance of basic life science and application to development of novel therapeutics. The cryoEM method with single particle analysis astonished to innovate the structure determination of biological macromolecules including this work. But I am afraid that the immature structure analysis reports may harmful around fields, because of the highly influential to them. Especially, ligand assignment and refinement is crucial endless work (Roversi, P., and Tronrud, D. E. (2021) Ten things I 'hate' about refinement. *Acta Crystallographica Section D Structural Biology* 77, 1497-1515).

I am a little happy that the major revised version is quite better than the original reviewed, especially on the ligand identification and ligand interaction description with the BLT1. Unfortunately the natural lipid receptor ligands including LTB4 for BLT1 have quite low affinities than the solved artificial ligand chemicals like BIIL260 (Ref-8) and MK-D-046 (Ref-9) and it is common situation in X-ray crystallographic structural studies due to be needed more harsh procedure to obtain diffraction quality crystals, in particular membrane proteins including GPCRs.

Especially, the ligand assignment of lipids and its refinement is tough work in structure biology for unexpected bound ligand(s) (Tilley, S. J., Skippen, A., Murray-Rust, J., Swigart, P. M., Stewart, A., Morgan, C. P., Cockcroft, S., and McDonald, N. Q. (2004) Structure-function analysis of human phosphatidylinositol transfer protein alpha bound to phosphatidylinositol. *Structure* 12, 317-326). So I had reviewed to comment based on my stringent standard, because I believe it is more constructive both the readers and authors of the structure paper.

Frankly speaking, yet I am not fully agreed with the structure modeling at this point, since it is very hard to agree on the BLT4 and BLT1 orthosteric site binding mode with water mediated interaction. I hope the authors describe more on BLT1 and BLT2 as described in Introduction, for example, the different specificities of the intrinsic lipid ligands LTB4 and 12-HHT for these related GPCRs, since it is also important in practice of therapeutic development.

Yet I have some reviewer opinions on the revised manuscript as follows with many minor points in a separate file.

First of all, the experimental description is better than the original, but the structural determination evidence by cryo-EM is almost the same as original. I hope it could be improved to justify in view from scientific standard in the executed experimental processes, in particular, on the refinement, since I felt some description of structure determination is not mention in detail in comparison with recent publication in Nature communication, which is an open access journal.

For example, the cited original EM image photo as the first of Supplementary Fig. 2, I am afraid that, is too crowded to pick each particle for single particle alignment and superimposed them with a same pose with a same orientation.

I agree that the authors using structure comparison of the AlphaFold2 predicted structures of BLT2. But the citation code should be cited as AlphaFold code "AF-Q9NPC1-F1" based on my inspection of AlphaFold2 predicted GPCR structure accuracy.

Additional structure insight of the determined EM map by the authors has been improved with the chemical confirmation including by the additional experiment of LC-MS analysis of the lipid extract fraction in the purified BLT1 expressed sample in Sf-9. I am interested whether the PI bound site of BLT1 is specific for PI or not, it should correlated to K_a values of PI and PI-phosphates. And it results in cell-signaling different manner via BLT1 (Kuniyeda, K., Okuno, T., Terawaki, K., Miyano,

M., Yokomizo, T., and Shimizu, T. (2007) Identification of the intracellular region of the leukotriene B-4 receptor type 1 that is specifically involved in G(i) activation. *Journal of Biological Chemistry* 282, 3998-4006). I guess it may be the reserving site for PI-kinase, since the PI3,4,5triphosphate kinase exerts crucial for neutrophil migration accompanied with BLT4 as the ligand of BLT1 as you know. Furthermore, I hope for the authors to mention that there are lipid (or PE) at the same binding site of PI binding with tunnel(!!) in the MK-D-046-bound BLT1 (PDB:7k15), however the paper did not mention the binding PE molecule with their binding mode comparison.

Fig.2e could be swapped with Supplementary Fig. 6c, since SFig. 6 is more informative than Fig. 2e on the interaction to BLT1 side chains.

In Lines 271-300

The open lid of BLT1 with Fig. 5 and is already described previously in Ref. 8 and 9 in detail, and the situation of the LTB4 binding mode is quite similar to the inhibitor ones. Thus, it is not novel to describe in detail again with Fig. 5. It should be more concise in a sentence without Fig. 5 with the reference citation due to too redundant. And Fig. 5 can be exchanged by Supplementary Fig. 5 because the authors described in detail with reference to SFif5, and it could be more comprehensive.

I cited descriptions from these papers.

From Ref. 8

"The vestibule of BLT1 is open on the extracellular surface (Fig. 1f), and the bound BIIL260 was not visible from the membrane side (Supplementary Fig. 4b), supporting the proposal that the BLT1 ligands, including BIIL260 and LTB4, may enter and leave the ligand-binding site via the extracellular surface. By contrast, the vestibules of other lipid-ligand GPCRs, including GPR40 (ref. 29), S1P1R30, CB1 (ref. 31), and rhodopsin32, are not open on the extracellular surface."

From Ref. 9

"The extracellular regions also differ between hBLT1 and gpBLT1. While the orthosteric binding pocket of hBLT1 is widely exposed to the solvent at the extracellular side (Fig. 1b), that of gpBLT1 is partially blocked."

Supplementary Fig. 5a.

a, Surface color is more informative using electrostatic potential than hydrophobicity and hydrophilicity surface because it can imagine direct non-bonding interaction directly.

Many minor points as follows; see also edited PDF file as well,

Minor proof point suggestion

Minor proof suggested:

line

85 which has the S1163.51Y mutation also show enhanced

> which has the S1163.51Y mutation at the DRY motif also show enhanced

188 the outward displacement of TM6.

> the outward displacement of TM6 by the receptor activation.

190 pocket show that

> pocket shows that

199 there is a big swap of the side chain of M1013.36

> there is a χ -angle rotation of the side chain of M1013.36

200 the alkyl tail of LTB4 to the position

> the alkyl tail of LTB4 where is the position

201 the antagonist-bound receptor M1013.36.

> that of M1013.36 in the antagonist-bound receptor.

203 displacement. Interestingly, both

> displacement. Both

204 show a large drop of receptor activation

> significantly lose the receptor activity

205 antagonist binding is the engagement of polar

> antagonist binding is direct and indirect polar

206 MK-D-046 binding, R156 4.64 forms direct hydrogen bonds
> MK-D-046 binding, R156 4.64 forms hydrogen bonds
207 of MK-D-046 and H94 3.29 forms direct hydrogen bonds
> of MK-D-046 and H94 3.29 forms hydrogen bonds
209-210 Whereas in agonist LTB4 binding, water molecules act as bridges to connect R156 4.64 and H94 3.29 to LTB4
> Whereas agonist LTB4 binds with R156 4.64 and H94 3.29 via water molecules in hydrogen bond network
210
>
213-214 phenomena about the antagonists binding is a network of polar interaction in the deep lower part of ligand binding pocket, which seems to lock the receptor in inactive
> these antagonists penetrate direct into the deep hole of lower part in the center of seven transmembrane bundle with polar interaction with the receptor to keep in inactive
218 in numerous studies
> among numerous studies
220-223 M101 3.36 sways to the center, together with the inward displacement of I271 7.39 and the tilting of F275 7.43, totally inhibit the entry of the sodium ion from locking the lower part of pocket (Fig. 3d), thus allowing the receptor to break the lock and rearrange the lower core for the α H5 of Gai binding.
> M101 3.36 swing into the center, with the inward side-chain rotation of I271 7.39 accompanied with the tilting of F275 7.43, to shut-down the sodium ion to enter the lower part of pocket (Fig. 3d), thus allowing the receptor to unlock and rearrange the lower core for the α H5 of Gai binding.
225 in the inactive conformation.
> in inactive.
227 almost totally abridged receptor activation
> almost totally cripple receptor activation
248 NSTR1 G i interaction 23.
> NSTR1 G i interaction (Fig. 4b) 23.
251 carbonyl of L353 G.H5.25 of Gai .
> carbonyl of L353 G.H5.25 of Gai (Fig. 4a).
255 the Gai (Fig. 4c-d).
> the Gai (Fig. 4c and 4d).
292 the acryl head
> the carboxy head group
299-300 and this may partially explain why anti-leukotriene drug development succeeds in CysLT, but not BLT.
> and it may be a reason why anti-leukotriene drugs were developed successful in CysLT, but not BLT (some suitable review article should cite ex) Yokomizo, T., Nakamura, M., and Shimizu, T. (2018) Leukotriene receptors as potential therapeutic targets. *J Clin Invest* 128, 2691-2701
Shirasaki, H. (2008) Cysteinyl leukotriene receptor CysLT(1) as a novel therapeutic target for allergic rhinitis treatment. *EXPERT OPINION ON THERAPEUTIC TARGETS* 12, 415-423).
374 and 384 temperature of 4 0C.
> temperature of 4 °C.
385 on a FEI 200 kv
> on a FEI 200 kV
389 set to 20 ev.
> set to 20 eV.
404 sampssling to yield
> sampling to yield
415 Phenix real_space refinement
> Phenix real space refinement
426 as before.
> as described before.
472 Van der Waals interactions.
> van der Waals interactions.
480 and 481 dG
> Δ G
536 was aligned

> was superimposed

Reviewer #2 (Remarks to the Author):

As I said in my previous report, this work by Wang et al. represents an important contribution to the field as it describes for the first time an active conformation for this receptor in complement to the two inactive structures already described in the literature. In that revised version, the authors added additional experiments associated with in silico calculations to improve data analysis, in particular the interaction of the ligand in its orthosteric pocket and the identification and localization of two different lipid cofactors. 6 more co-authors have been added to the list regarding their respective contributions. While I think the mass spect. analysis is convincing and undeniably brings an added value to better identify the lipid cofactor(s), I still have doubts about the interaction of the ligand in its pocket.

Overall I think this is a nice study but, on my point of view, it deserves a better analysis and writing.

Additional comments:

1. About LTB4 and its interaction with the receptor. So as a major contribution to this revised version, the authors claim that they could identified a cluster of 5 water molecules including one that interacts with the OH in position 5 of the ligand. Even if I am not an expert in this, my major concern about this is how at a 2.91Å resolution could it be possible to identify water molecules? My background in biophysics tells me that we need a much better resolution to get this. And this is not described in the revised manuscript, i.e. how they could observe experimentally these molecules (we just have that additional sentence on line 122: "we carefully examine the density of the pocket...". I think MD simulations can be helpful and relevant on the basis of strong experimental evidences. I am not sure this is the case concerning these water molecules or I missed something somewhere.

There is still no interactions observed with the other OH group which is close to E185 and also no interaction(s) involving the acidic function of the eicosanoid while we know that without these two functions the affinity is much lower to BLT1. This should be discussed somewhere in the manuscript, i.e. just one OH group interacting with a putative water molecule associated with a bench of van der waals interactions, this sounds pretty weak for me.

line 140: it misses the unity: I guess this is kcal/mol? That difference of 0.3 kcal/mol does that make the difference, i.e. to justify at least one molecule of water bridging an interaction between the ligand and the receptor? In addition, what is the relevancy of a DeltaG based on in silico calculations relatively to a difference of 0.3 kcal/mol? It misses a comment on that in the manuscript.

2. SRE exp. The description of these experiments starts on line 142 without any address to a Fig. Concerning these experiments and that § (lines 142 to 157), I am not sure that concerning the residues that impact the most the activity that the relative differences between them are meaningful. For instance, when it is stated on line 153 that the "greatest decrease of activity is seen on the polar mutant...".

3. About the lipid cofactor. By contrast with the analysis of ligand/receptor interaction, this part corresponds to a real improvement in this revised version. I was wondering why the second identified lipid (PI 18:0/18:1) was discarded in the analysis as its occupancy based on mass spec analysis is not so different from the other one? What would mean the presence of two different lipids?

4. The authors report additional directed-mutagenesis exp associated with SRE assays in Fig. S6d. Based on these experiments, how could we assess that this is the absence of the lipid that would be responsible of this decrease in activity of the receptor observed? As I said in my previous

report, I think the methods is not described enough for this assay. And probably this is not the right assay to prove the allosteric effect of such a lipid as we need at least a negative control.

Minor points :

- I still think the introduction could have been better regarding BLT signaling.
- Have the authors checked if D114 is a conserved residue in helix III in BLT receptors or in other GPCRs in general? Also, this could help the reader to clearly represent D114 and R129 in Fig. 2e.
- The end of the § dedicated to the "Allosteric site of PI binding" is just based on x,y,z considerations. We would expect a deeper analysis instead of just a simple comparison with the HT1A receptor.
- just a simple curiosity. About the upper open pocket of BLT1, I was wondering if at equilibrium there is significant differences in Kd values (or even better, i.e. koff values) between agonist that interacts with a GPCR that positions a lipid or not above the orthosteric pocket.

Reviewer #3 (Remarks to the Author):

The authors have made satisfactory corrections to the manuscript based on my comments.

One further change is required. There is no biochemical data to show that PI binding to the receptor alters the affinity of either agonists or antagonists and therefore it cannot be concluded that it is an 'allosteric' ligand. All mention of 'allosteric' in relation to the PI lipid binding pocket must be deleted.

Reviewer #4 (Remarks to the Author):

Review: NCOMMS-21-39346A

In this work, the authors have reported the cryo-electron microscopy structure of LTB4 -bound human BLT1 in complex with a Gi protein in an active conformation. This manuscript is important to understand the structural basis of leukotriene B4 receptor activation as well as to design the anti-leukotriene drugs. The novelty of this study is that the bound complex reported here seem reasonable. The points raised by the reviewer 1 and 2 have been adequately addressed. I would recommend publication of this study given that minor amendments are made to the text.

- 1) In the methods section authors should mention the free energy analysis method that was used to score the active site in terms of the binding free energy.
- 2) The choice of solute and solvent dielectric constant used in the free energy calculations, and the rationale for the choice of solute dielectric constant is also need to be stated.

REVIEWER COMMENTS

Reviewer #1 (Remarks to the Author):

The major revised version is the many improvements of BLT1 complex structure with Gi with cryoEM in the various features according to the other reviews including my opinion, and the quality reached to meet to publish in a well-known journal like Nature communication with obvious importance of basic life science and application to development of novel therapeutics. The cryoEM method with single particle analysis astonished to innovate the structure determination of biological macromolecules including this work. But I am afraid that the immature structure analysis reports may harmful around fields, because of the highly influential to them. Especially, ligand assignment and refinement is crucial endless work (Roversi, P., and Tronrud, D. E. (2021) Ten things I 'hate' about refinement. Acta Crystallographica Section D Structural Biology 77, 1497-1515).

Through a thorough analysis of our structure and the density map, we have identified a direct interaction between the carboxyl head of the BLT4 and N268, as well as two new water molecules that connect to the carboxyl group of BLT4, and those interactions enhance the hydrogen network that lock the ligand in the pocket (Fig 2a).

I am a little happy that the major revised version is quite better than the original reviewed, especially on the ligand identification and ligand interaction description with the BLT1. Unfortunately the natural lipid receptor ligands including LTB4 for BLT1 have quite low affinities than the solved artificial ligand chemicals like BIIL260 (Ref-8) and MK-D-046 (Ref-9) and it is common situation in X-ray crystallographic structural studies due to be needed more harsh procedure to obtain diffraction quality crystals, in particular membrane proteins including GPCRs.

Especially, the ligand assignment of lipids and its refinement is tough work in structure biology for unexpected bound ligand(s) (Tilley, S. J., Skippen, A., Murray-Rust, J., Swigart, P. M., Stewart, A., Morgan, C. P., Cockcroft, S., and McDonald, N. Q. (2004) Structure-function analysis of human phosphatidylinositol transfer protein alpha bound to phosphatidylinositol. Structure 12, 317-326). So I had reviewed to comment based on my stringent standard, because I believe it is more constructive both the readers and authors of the structure paper.

Frankly speaking, yet I am not fully agreed with the structure modeling at this point, since it is very hard to agree on the BLT4 and BLT1 orthosteric site binding mode with water mediated interaction. I hope the authors describe more on BLT1 and BLT2 as described in Introduction, for example, the different specificities of the intrinsic lipid ligands LTB4 and 12-HHT for these related GPCRs, since it is also important in practice of therapeutic development.

As mentioned early, we have identified a direct interaction between the LTB4 c1 carboxyl group and the N268 which lock the head of LTB4 in the pocket. We also added more background information of BLT1 and BLT2 in the introduction part and compare the chemical difference

between LTb4 and 12-HHT (lack of the middle C5 hydroxyl group) and discuss the specificity in the main text.

Yet I have some reviewer opinions on the revised manuscript as follows with many minor points in a separate file.

First of all, the experimental description is better than the original, but the structural determination evidence by cryo-EM is almost the same as original. I hope it could be improved to justify in view from scientific standard in the executed experimental processes, in particular, on the refinement, since I felt some description of structure determination is not mention in detail in comparison with recent publication in Nature communication, which is an open access journal.

We have added more information in the data processing part.

For example, the cited original EM image photo as the first of Supplementary Fig. 2, I am afraid that, is too crowded to pick each particle for single particle alignment and superimposed them with a same pose with a same orientation.

Now, it is common and well accepted that high particle density leads higher resolution, as higher particle density generally correlates with higher signal/noise ratio, which is critical for getting high resolution, for instance, recent GPCR cryoEM structure Danev et al. Nat Comm DOI: 10.1038/s41467-021-24650-3; Liang et al. ACS Pharmacol. Transl. Sci. 2020, 3, 263–284. The updated cryolo, relion and cryosparcm program can all well recognize and handle crowded particle at a very high density.

I agree that the authors using structure comparison of the AlphaFold2 predicted structures of BLT2. But the citation code should be cited as AlphaFold code "AF-Q9NPC1-F1" based on my inspection of AlphaFold2 predicted GPCR structure accuracy.

Done.

Additional structure insight of the determined EM map by the authors has been improved with the chemical confirmation including by the additional experiment of LC-MS analysis of the lipid extract fraction in the purified BLT1 expressed sample in Sf-9. I am interested whether the PI bound site of BLT1 is specific for PI or not, it should corelated to Ka values of PI and PI-phosphates. And it results in cell-signaling different manner via BLT1 (Kuniyeda, K., Okuno, T., Terawaki, K., Miyano, M., Yokomizo, T., and Shimizu, T. (2007) Identification of the intracellular region of the leukotriene B-4 receptor type 1 that is specifically involved in G(i) activation. Journal of Biological Chemistry 282, 3998-4006). I guess it may be the reserving site for PI-kinase, since the PI3,4,5triphosphate kinase exerts crucial for neutrophil migration accompanied with BLT4 as the ligand of BLT1 as you know. Furthermore, I hope for the authors to mention that there are lipid (or PE) at the same binding site of PI binding with tunnel(!!) in the MK-D-046-bound BLT1 (PDB:7k15), however the paper did not mention the binding PE molecule with their binding mode comparison.

We have examined the pocket in detail; we agree with you that this could be a reserving site for PI-kinase, as we mentioned early, this pocket can easily accommodate PtdIns4P (Fig. S2f). Although, we did not observe a corresponding PE peak in our LC-MS analysis, we agree with you that in real-membrane environment, PE molecule as component of membrane, can utilize this pocket. We therefore model an PE 16:0/16:0 molecule, one of the most common PEs in mammalian membrane, into the pocket, interestingly, we found the amine head of PE can be nicely fitted into the tunnel discovered in the MK-D-046-bound BLT1 crystal structure, where a nonaethylene glycol (P2E) molecule snuggles in (Fig. S6g). We have discussed this in the main text.

Fig.2e could be swapped with Supplementary Fig. 6c, since SFig. 6 is more informative than Fig. 2e on the interaction to BLT1 side chains.

Done.

In Lines 271-300

The open lid of BLT1 with Fig. 5 and is already described previously in Ref. 8 and 9 in detail, and the situation of the LTB4 binding mode is quite similar to the inhibitor ones. Thus, it is not novel to describe in detail again with Fig. 5. It should be more concise in a sentence without Fig. 5 with the reference citation due to too redundant. And Fig. 5 can be exchanged by Supplementary Fig. 5 because the authors described in detail with reference to SFif5, and it could be more comprehensive.

I cited descriptions from these papers.

From Ref. 8

“The vestibule of BLT1 is open on the extracellular surface (Fig. 1f), and the bound BIIL260 was not visible from the membrane side (Supplementary Fig. 4b), supporting the proposal that the BLT1 ligands, including BIIL260 and LTB4, may enter and leave the ligand-binding site via the extracellular surface. By contrast, the vestibules of other lipid-ligand GPCRs, including GPR40 (ref. 29), S1P1R30, CB1 (ref. 31), and rhodopsin32, are not open on the extracellular surface.”

From Ref. 9

“The extracellular regions also differ between hBLT1 and gpBLT1. While the orthosteric binding pocket of hBLT1 is widely exposed to the solvent at the extracellular side (Fig. 1b), that of gpBLT1 is partially blocked.”

Thank you for your suggestion. We have move Fig 5 to Fig. S9. And add the reference.

Supplementary Fig. 5a.

a, Surface color is more informative using electrostatic potential than hydrophobicity and hydrophilicity surface because it can imagine direct non-bonding interaction directly.

To be more informative, we put both hydrophobicity and electrostatic potential on Fig. s5a.

Many minor points as follows; see also edited PDF file as well,

Minor proof point suggestion

Minor proof suggested:

line

85 which has the S1163.51Y mutation also show enhanced

> which has the S1163.51Y mutation at the DRY motif also show enhanced

done

188 the outward displacement of TM6.

> the outward displacement of TM6 by the receptor activation.

done

190 pocket show that

> pocket shows that

done, thank you.

199 there is a big swap of the side chain of M1013.36

> there is a χ -angle rotation of the side chain of M1013.36

done.

200 the alkyl tail of LTB4 to the position

> the alkyl tail of LTB4 where is the position

done, thank you.

201 the antagonist-bound receptor M1013.36.

> that of M1013.36 in the antagonist-bound receptor.

done.

203 displacement. Interestingly, both

> displacement. Both

done.

204 show a large drop of receptor activation

> significantly lose the receptor activity

done.

205 antagonist binding is the engagement of polar

> antagonist binding is direct and indirect polar

done

206 MK-D-046 binding, R156 4.64 forms direct hydrogen bonds

> MK-D-046 binding, R156 4.64 forms hydrogen bonds

done

207 of MK-D-046 and H94 3.29 forms direct hydrogen bonds

> of MK-D-046 and H94 3.29 forms hydrogen bonds

done

209-210 Whereas in agonist LTB4 binding, water molecules act as bridges to connect R156 4.64 and H94 3.29 to LTB4

> Whereas agonist LTB4 binds with R156 4.64 and H94 3.29 via water molecules in hydrogen bond network

done, thank you.

210

>

213-214 phenomena about the antagonists binding is a network of polar interaction in the deep lower part of ligand binding pocket, which seems to lock the receptor in inactive

> these antagonists penetrate direct into the deep hole of lower part in the center of seven transmembrane bundle with polar interaction with the receptor to keep in inactive

done, thank you.

218 in numerous studies

> among numerous studies

done.

220-223 M101 3.36 sways to the center, together with the inward displacement of I271 7.39 and the tilting of F275 7.43, totally inhibit the entry of the sodium ion from locking the lower part of pocket (Fig. 3d), thus allowing the receptor to break the lock and rearrange the lower core for the α H5 of G α i binding.

> M101 3.36 swing into the center, with the inward side-chain rotation of I271 7.39 accompanied with the tilting of F275 7.43, to shut-down the sodium ion to enter the lower part of pocket (Fig. 3d), thus allowing the receptor to unlock and rearrange the lower core for the α H5 of G α i binding.

done, thank you.

225 in the inactive conformation.

> in inactive.

done

227 almost totally abridged receptor activation

> almost totally cripple receptor activation

done

248 NSTR1 G i interaction 23.

> NSTR1 G i interaction (Fig. 4b) 23.

done

251 carbonyl of L353 G.H5.25 of G α i .

> carbonyl of L353 G.H5.25 of G α i (Fig. 4a).

done.

255 the G α i (Fig. 4c-d).

> the G α i (Fig. 4c and 4d).

done

292 the acryl head
> the carboxy head group

done

299-300 and this may partially explain why anti-leukotriene drug development succeeds in CysLT, but not BLT.

> and it may be a reason why anti-leukotriene drugs were developed successful in CysLT, but not BLT (some suitable review article should cite ex) Yokomizo, T., Nakamura, M., and Shimizu, T. (2018) Leukotriene receptors as potential therapeutic targets. J Clin Invest 128, 2691-2701
Shirasaki, H. (2008) Cysteinyl leukotriene receptor CysLT(1) as a novel therapeutic target for allergic rhinitis treatment. EXPERT OPINION ON THERAPEUTIC TARGETS 12, 415-423).

Done.

374 and 384 temperature of 4 0C.
> temperature of 4 °C.

done

385 on a FEI 200 kv
> on a FEI 200 kV

done

389 set to 20 ev.
> set to 20 eV.

done

404 sampssling to yield
> sampling to yield

done, thank you.

415 Phenix real_space refinement
> Phenix real space refinement

done

426 as before.
> as described before.

Done.

472 Van der Waals interactions.
> van der Waals interactions.

done

480 and 481 dG
> ΔG

done

536 was aligned
> was superimposed

done

Reviewer #2 (Remarks to the Author):

As I said in my previous report, this work by Wang et al. represents an important contribution to the field as it describes for the first time an active conformation for this receptor in complement to the two inactive structures already described in the literature. In that revised version, the authors added additional experiments associated with in silico calculations to improve data analysis, in particular the interaction of the ligand in its orthosteric pocket and the identification and localization of two different lipid cofactors. 6 more co-authors have been added to the list regarding their respective contributions. While I think the mass spect. analysis is convincing and undeniably brings an added value to better identify the lipid cofactor(s), I still have doubts about the interaction of the ligand in its pocket.

Overall I think this is a nice study but, on my point of view, it deserves a better analysis and writing.

Additional comments:

1. About LTB4 and its interaction with the receptor. So as a major contribution to this revised version, the authors claim that they could identified a cluster of 5 water molecules including one that interacts with the OH in position 5 of the ligand. Even if I am not an expert in this, my major concern about this is how at a 2.91Å resolution could it be possible to identify water molecules? My background in biophysics tells me that we need a much better resolution to get this. And this is not described in the revised manuscript, i.e. how they could observe experimentally these molecules (we just have that additional sentence on line 122: "we carefully examine the density of the pocket...". I think MD simulations can be helpful and relevant on the basis of strong experimental evidences. I am not sure this is the case concerning these water molecules or I missed something somewhere.

Water molecules can be identified in cryo-EM map when the resolution lower than or around 3.0 Å using Gatan K2 Summit camera, please see the link (www.gatan.com/cn/resources/media-library/image-sample-data). And in fact many GPCR cryo-EM structures report the exist of water molecules in the ligand binding pocket at resolution around 3.0 Å, for instance, Apo 5-HT1A (pdb 7e2x) 3.0Å 9 water molecules in the pocket (Nature 2021, 10.1038/s41586-021-03376-8); EP2, 7cx2, 2.8Å, one water in the pocket (Sci. Adv. 2021; 7 : eabf1268).

There is still no interactions observed with the other OH group which is close to E185 and also no interaction(s) involving the acidic function of the eicosanoid while we know that without these two functions the affinity is much lower to BLT1. This should be discussed somewhere in the manuscript, i.e. just one OH group interacting with a putative water molecule associated with a bench of van der waals interactions, this sounds pretty weak for me.

Through a thorough examination of our structure and the density map, we have identified a direct interaction between the carboxyl group of BLT4 head and N268 (Fig. 2a) and an additional

cluster of water molecules, together with the previous identified hydrogen network around the C5 hydroxyl group, formed anchor LTB4 in the ligand binding pocket. In addition, our MD simulation study suggest that a hydrogen-bond network is also formed around the C12 hydroxyl group of LTB4 (Fig. s5b).

line 140: it misses the unity: I guess this is kcal/mol? That difference of 0.3 kcal/mol does that make the difference, i.e. to justify at least one molecule of water bridging an interaction between the ligand and the receptor? In addition, what is the relevancy of a ΔG based on in silico calculations relatively to a difference of 0.3 kcal/mol? It misses a comment on that in the manuscript.

Yes, the program uses kcal/mol as units for energy change calculation. However, this does not mean the energy change calculated in the program equal to real-world kcal/mol, it is just for comparison. Different programs use different methods to calculate the energy change and yet the accurate estimation of energy change is still a challenge in the field. To avoid misunderstanding, the ΔG has been replaced by 'docking score' in the main text. We think a difference of 0.3 docking score is a notable difference.

2. SRE exp. The description of these experiments starts on line 142 without any address to a Fig. Concerning these experiments and that § (lines 142 to 157), I am not sure that concerning the residues that impact the most the activity that the relative differences between them are meaningful. For instance, when it is stated on line 153 that the "greatest decrease of activity is seen on the polar mutant...".

We have changed the sentence to "dramatic decrease of ..." to avoid a quantitative comparison.

3. About the lipid cofactor. By contrast with the analysis of ligand/receptor interaction, this part corresponds to a real improvement in this revised version. I was wondering why the second identified lipid (PI 18:0/18:1) was discarded in the analysis as its occupancy based on mass spec analysis is not so different from the other one? What would mean the presence of two different lipids?

We agree with you. Ideally, we should build two models. However, the only difference between PI 18:0/16:1 and PI 18:0/18:1 is the alky chain length. And the weak resolution on the alky chain prevents us to distinguishing PI 18:0/16:1 and PI 18:0/18:1 from current map. In principle, one map should correspond to one model. We therefore build PI 18:0/16:1 as a representative model based on its higher occupancy. To rule out the possibility of a large conformational difference of PI 18:0/18:1 in the pocket, we have also built a model for the second PI (18:0/18:1) and compare it with PI 18:0/16:1 (Fig s6c). We found they are almost identical except the length of the alky tail. We have discussed this in the main text.

4. The authors report additional directed-mutagenesis exp associated with SRE assays in Fig. S6d. Based on these experiments, how could we assess that this is the absence of the lipid that would be responsible of this decrease in activity of the receptor observed? As I said in my previous

report, I think the methods is not described enough for this assay. And probably this is not the right assay to prove the allosteric effect of such a lipid as we need at least a negative control.

Thank you for your comment. We agree that the SRE reporter assay may not be the perfect assay for this study. However, we think it is an effective assay to detect BLT1 Gi activity based on the consistency of the key residue mutation (e.g. H94A, R156A, R267A ...) with the structural observation, and the consistency with previous report (ref 15). We think in this case (Fig. S6d, the mutation of the PI binding site), the issue is not the assay, it is the lack of the proposed lipid (due to the difficulty of obtaining PI 18:0/16:1 or 18:0/18:1) in the system. As suggested by reviewer #1, we have discussed the possibility of other molecules binding (PIP, PE or other phosphate group) and successfully modeled them in the pocket (Fig. S6g). Also, our initial data (first submission) has shown other molecule like PtdIns4P can positively regulate the receptor activity, suggesting that other molecules may utilize this binding site to regulate receptor activity. We have discussed this in the main text in responding to reviewer # 1's comment.

Minor points :

- I still think the introduction could have been better regarding BLT signaling.

We have added more information of BLT signaling in the introduction part, especially a comparison of BLT1 and BLT2 ligands and discussed the specificity in the discussion.

-Have the authors checked if D114 is a conserved residue in helix III in BLT receptors or in other GPCRs in general? Also, this could help the reader to clearly represent D114 and R129 in Fig. 2e.

It is conserved in lipids receptor except CysLT1,2, Fig. s2e, we have discussed this in main text.

- The end of the § dedicated to the "Allosteric site of PI binding" is just based on x,y,z considerations. We would expect a deeper analysis instead of just a simple comparison with the HT1A receptor.

Thank you for your suggestion. To be stringent, we have delete the word of "allosteric". We agree that although LC-MS did not identify other lipid molecule, the PI binding site is not a site only for PI binding, it may also provide a pocket for other molecules in the real membrane environment, for instance, it may serve as an PI-kinase docking site as the PI3,4,5triphosphate kinase play crucial role for neutrophil migration upon LTB4 stimulation. It can also serve a binding site for PE binding. For this purpose, we have successfully modelled PIP and PE (16:0/16:0) in the pocket, and interestingly we found PE may use the same tunnel as discovered in the MK-D-046-bound BLT1 crystal structure paper (Fig. s6g). Those data suggest the versatility of this site. We have discussed this in detail in the main text.

- just a simple curiosity. About the upper open pocket of BLT1, I was wondering if at equilibrium there is significative differences in Kd values (or even better, i.e. koff values) between agonist that interacts with a GPCR that positions a lid or not above the orthosteric pocket.

It is difficult to accurately compare the binding kinetic of ligand binding with lid on or not, as ligand shape and receptor difference all influence the outcome. However, we find the AM11542 which interact intensively with the lid of CB1 (Fig. s9d) has significant lower EC50 than AM841 (3.5 nM vs 13.8 nM) which only slightly touch the lid (Nature 2017, DOI doi:10.1038/nature23272), suggesting that the lid binding make a difference.

Reviewer #3 (Remarks to the Author):

The authors have made satisfactory corrections to the manuscript based on my comments.

One further change is required. There is no biochemical data to show that PI binding to the receptor alters the affinity of either agonists or antagonists and therefore it cannot be concluded that it is an 'allosteric' ligand. All mention of 'allosteric' in relation to the PI lipid binding pocket must be deleted.

Thank you for your suggestion. We agree with you that without strong biochemical data the word of allosteric is meaningless. We have deleted the word "allosteric". However, as suggested by review # 1, we discussed the possibility of this site to serve as additional docking site for other molecules, such as PIP and PE. And we have successfully modeled those molecules into the pocket (Fig. s6g), suggesting a versatility of the site. The real function of this site need to be further investigate in the future.

Reviewer #4 (Remarks to the Author):

Review: NCOMMS-21-39346A

In this work, the authors have reported the cryo-electron microscopy structure of LTB4 -bound human BLT1 in complex with a Gi protein in an active conformation. This manuscript is important to understand the structural basis of leukotriene B4 receptor activation as well as to design the anti-leukotriene drugs. The novelty of this study is that the bound complex reported here seem reasonable. The points raised by the reviewer 1 and 2 have been adequately addressed. I would recommend publication of this study given that minor amendments are made to the text.

1) In the methods section authors should mention the free energy analysis method that was used to score the active site in terms of the binding free energy.

We did not analyze the binding free energy as our focus is the stability (position) of the ligand in the pocket. Rather, we have checked the pose and position of the ligand in the MD simulation and used molecular docking to confirm the binding pose. In our revised manuscript, the $\Delta\Delta G$ has been replaced by 'docking score' to avoid misunderstanding. We agree with you that a free energy analysis may provide a more precise evaluation of ligand. However, as current free energy calculation (MMPB/GBSA, FEP, or TI) are normally applied for the comparison between different ligands and not-well optimized for membrane proteins, we did not attempt the energy calculation algorithms for our pose confirmation task.

2) The choice of solute and solvent di-electric constant used in the free energy calculations, and the rationale for the choice of solute di-electric constant is also need to be stated.

In our systems, parameters for solute were defined by FF19SB for protein and GAFF2 for ligand. For solvents, LIPID17, FF19SB force field, and TIP3P model were used for POPC and cholesterol membrane, KCl, and water, respectively. Since we did not do the free energy calculation, we did not specify the di-electric constant for solute and solvent in the analysis.

Peer review comments, third round review - -

Reviewer #1 (Remarks to the Author):

The cryoEM structure of the activated BLT1 complexed LTB4 and lipid PI and some others using sophisticated constructs shows a few novel important features including LTB4 binding mode and a novel allosteric binding site of PI where could involve possible allosteric or additional activating pathway for PI4P signaling directly (ref. 18. 19) in the second round revised manuscript, it is sound and meets to highly scientific standard with novel and original by well experimental supports and well description.

A few possible revision points as follows:

280-282 described the hydrogen bond interaction of -NH of G2907.58 and C=O of E28 G.HN.52 of the α N of Gai, but no label indication of E28G.HN.52 in Fig 4a, whereas R2186.32 was labeled without any mention in text.

Minor point

101 was purified from sf9 cells -> was purified from Sf9 cells

Reviewer #2 (Remarks to the Author):

First of all, I would like to address my sincere congratulations to this work. I think this is an important contribution to the field and I wish this will help to define new interesting therapeutic compounds in the future. In the following I have just some suggestions. if some can be useful I would be happy.

At 2.91 Å resolution this is still difficult to affirm with certainty that some pieces of electron densities do correspond to water molecules. It may be water, but I would suggest to use the conditional mode concerning the possibility it could be H₂O or to at least formulate a kind of self-criticism on that particular point. As stated in the paper mentioned by reviewer 1 (Roversi & Tronrud 2021), "refinement never ends" so one should be careful at such resolution, but I understand that this would not be an argument to further delay the publication. Again, I am not a specialist in structure calculations from electron density maps, but at ~3Å, in absolute terms, it could be some noise or something else chosen as part of a larger group of atoms of which we see that part as the tip of an iceberg, it could also be an ion. If we see water at this resolution (and in some cases, it is possible), it must be well anchored -as suggested in the model presented herein. Interactions with nearby atoms must be compatible with hydrogen bonds, nature of the group and distance (typically 2.8 - 3.2 Å dist with O), but if the distance is too short (many people often forget this criterion too much) it suggests rather the presence of an ion or a covalent fragment with a density interrupted. Could it be possible to indicate in Fig. 2a the length of the hydrogen bonds between these putative water molecules with the ligand or the receptor?

The authors mentioned also a newly detected direct interaction between the LTB4 acid function and the residue N268, residue that once mutated has no impact on BLT1 activity (Fig. 2d). I would have also added a comment on this and, beyond, I would have tried to put in perspective the present interactions between the LTB4 and BLT1 with what we know about the binding of various different eicosanoids to BLT1, which is well documented in the literature. For instance, the fact that the mutation of N268, which is close to COOH-LTB4, has no impact on the activity of BLT1 is in accordance with the literature based on the study of the interaction of LTB4 and structurally related eicosanoids with BLT1. Furthermore, the LTB4 modified with the fluorophore Alexa Fluor 568 at the acid function end can still bind to and activate BLT1. This strongly suggests that the acid function is not mandatory for the LTB4 to interact with BLT1.

Again, these are just suggestions. Congratulations for this work.

Reviewer #4 (Remarks to the Author):

The authors have adequately addressed all my comments.

REVIEWERS' COMMENTS

Reviewer #1 (Remarks to the Author):

The cryoEM structure of the activated BLT1 complexed LTB4 and lipid PI and some others using sophisticated constructs shows a few novel important features including LTB4 binding mode and a novel allosteric binding site of PI where could involve possible allosteric or additional activating pathway for PI4P signaling directly (ref. 18. 19) in the second round revised manuscript, it is sound and meets to highly scientific standard with novel and original by well experimental supports and well description.

Thank very much for your positive comments on our study!

A few possible revision points as follows:

280-282 described the hydrogen bond interaction of -NH of G2907.58 and C=O of E28 G.HN.52 of the α N of G α i, but no label indication of E28G.HN.52 in Fig 4a, whereas R2186.32 was labeled without any mention in text.

Thank you for your comment, we have added the missing label in Fig. 4a. also the missing description of the R218/C351 interaction in the main text.

Minor point

101 was purified from sf9 cells -> was purified from Sf9 cells

Fixed, thank you.

Reviewer #2 (Remarks to the Author):

First of all, I would like to address my sincere congratulations to this work. I think this is an important contribution to the field and I wish this will help to define new interesting therapeutic compounds in the future. In the following I have just some suggestions. if some can be useful I would be happy.

Thank you very much for your comments and suggestions!

At 2.91 Å resolution this is still difficult to affirm with certainty that some pieces of electron densities do correspond to water molecules. It may be water, but I would suggest to use the conditional mode concerning the possibility it could be H₂O or to at least formulate a kind of self-criticism on that particular point. As stated in the paper mentioned by reviewer 1 (Roversi & Tronrud 2021), “refinement never ends” so one should be careful at such resolution, but I understand that this would not be an argument to further delay the publication. Again, I am not a specialist in structure calculations from electron density maps, but at ~3Å, in absolute terms, it could be some noise or something else chosen as part of a larger group of atoms of which we see that part as the tip of an iceberg, it could also be an ion. If we see water at this resolution (and in some cases, it is possible), it must be well anchored -as suggested in the model presented herein. Interactions with nearby atoms must be compatible with hydrogen bonds, nature of the group and distance (typically 2.8

- 3.2 Å dist with O), but if the distance is too short (many people often forget this criterion too much) it suggests rather the presence of an ion or a covalent fragment with a density interrupted. Could it be possible to indicate in Fig. 2a the length of the hydrogen bonds between these putative water molecules with the ligand or the receptor?

We agree with you that we are not absolutely sure that those densities correspond to water molecules at current resolution (2.9Å). However, given the factors that water molecules bridge receptor and LTB4 throughout the whole MD simulations, and the docking data shows that water molecules “significantly” (to be more scientific, notably) lower the free energy, we think those molecules are highly likely to be water molecules. We also measured the distance between water and the connected residues and LTB4. The distances vary from 3.0Å to 3.3Å (Fig 2A), in the good range of forming hydrogen bond. We have modified the description to reflect these factors in the main text. Thank you for your suggestion.

The authors mentioned also a newly detected direct interaction between the LTB4 acid function and the residue N268, residue that once mutated has no impact on BLT1 activity (Fig. 2d). I would have also added a comment on this and, beyond, I would have tried to put in perspective the present interactions between the LTB4 and BLT1 with what we know about the binding of various different eicosanoids to BLT1, which is well documented in the literature. For instance, the fact that the mutation of N268, which is close to COOH-LTB4, has no impact on the activity of BLT1 is in accordance with the literature based on the study of the interaction of LTB4 and structurally related eicosanoids with BLT1. Furthermore, the LTB4 modified with the fluorophore Alexa Fluor 568 at the acid function end can still bind to and activate BLT1. This strongly suggests that the acid function is not mandatory for the LTB4 to interact with BLT1.

Thank you very much for your suggestion! Yes, the mutagenesis data shows that the role of N268 is not determinant, may be only additive. We are not very surprised about the data as our MD data shows that there is some degree of freedom on the carboxyl group of LTB4 (movie 1), also a snapshot analysis of the MD shows that the carboxyl group forms a H-bond with the backbone of TM7, and there is a vast water network interaction around the carboxyl end of LTB4 (Fig s5b). More importantly, as you suggested, the carboxyl end modified fluorescent LTB4s are full agonist of BLT1 (Sabirsh et al, Journal of Lipid Research 2005). Thank you for your suggestion which enhances the quality of our paper. We have added a discussion of the N268 mutation in our main text.

Again, these are just suggestions. Congratulations for this work.

Reviewer #4 (Remarks to the Author):

The authors have adequately addressed all my comments.

Thank you.